# MaxCutBench: Revisiting and Benchmarking Graph Neural Networks for Maximum Cut

**Ankur Nath**                                                                          *anath@tamu.edu*
*Department of Computer Science & Engineering*
*Texas A&M University*

**Alan Kuhnle**                                                                         *kuhnle@tamu.edu*
*Department of Computer Science & Engineering*
*Texas A&M University*

**Reviewed on OpenReview:** *https://openreview.net/forum?id=322PpCGAX8*

## Abstract

Recently, there has been much work on designing general heuristics for graph-based, combinatorial optimization problems via the incorporation of Graph Neural Networks (GNNs) to learn distribution-specific solution structures. However, there is a lack of consistency in evaluating these heuristics in terms of the baselines and instances chosen, making it difficult to assess the relative performance of the algorithms. In this paper, we introduce **MaxCutBench**—an open-source benchmark suite dedicated to the NP-hard Maximum Cut problem. The suite offers a unified interface for 16 algorithms, both traditional and machine-learning-based. Using our benchmark, we conduct an in-depth analysis of the implemented algorithms on a carefully selected set of hard instances from diverse graph datasets. Our main finding is that classical local search heuristics can outperform several highly cited learning-based approaches, including S2V-DQN (Khalil et al., 2017), ECO-DQN (Barrett et al., 2020), among others, in terms of objective value, generalization, inference time, and scalability. Additionally, we find that the performance of ECO-DQN either remains the same or improves when the GNN is replaced by simple linear regression. We hope our benchmark will contribute to the efforts of the community to standardize the evaluation of learned heuristics for combinatorial optimization. Code, data, and pre-trained models are available at: `https://github.com/ankurnath/MaxCut-Bench`.

## 1 Introduction

The design of effective heuristics or approximation algorithms for NP-hard combinatorial optimization (CO) problems is a challenging task, often requiring domain-specific knowledge alongside a rigorous process of empirical refinement. Typically, the precise probability distribution of a particular set of instances that are needed for a given application is complex or unknown and may deviate far from the set of worst-case instances that give rise to the computational complexity of the problem at hand. For example, consider a shipping company that must solve a presumably similar optimization problem each day for the routing of its delivery vehicles. Consequently, there has been significant interest among researchers in automating this demanding and tedious design process using machine learning to develop algorithms that exploit the inherent structure of these distributions (Khalil et al., 2017; Barrett et al., 2020; 2022; Zhang et al., 2023; Tönshoff et al., 2022). Empirical evidence suggests that learned heuristics (Barrett et al., 2020; Tönshoff et al., 2022) can be competitive with state-of-the-art (SOTA) heuristics tailored to individual problems.

However, along with the surge of automated, general heuristics for CO problems, it is necessary to have a standardized way to evaluate these heuristics to determine what gains, if any, are achieved over more traditional heuristics. The current status of the field is that each work formulates a heuristic and then selects its own baselines, problems, and instance distributions to evaluate the heuristic. Often, a combination of weak traditional heuristics and expensive exact algorithms is employed as baselines, such as the Greedy

algorithm (weak) or exact IP solvers such as GUROBI (Gurobi Optimization, LLC, 2023) and CPLEX (IBM-ILOG, 2023). More discussion and details can be found in Appendix A.3.

In this paper, we analyze several highly cited and recently proposed learned heuristics for the Maximum Cut (MaxCut) problem. MaxCut is one of the fundamental $\mathcal{NP}$-hard optimization problems (Karp, 1972) and has been intensely studied for decades (Darvariu et al., 2024). It has numerous real-world applications, such as portfolio optimization (Venturelli & Kondratyev, 2019; Elsokkary et al., 2017), protein folding (Perdomo-Ortiz et al., 2012), VLSI design (Barahona et al., 1988), and finding the ground state of the Ising Hamiltonian in physics (Barahona, 1982).[1] Significant commercial and research efforts have been devoted to developing MaxCut solvers using classical (Goto et al., 2019) and quantum annealing (Leleu et al., 2019; Tiunov et al., 2019) approaches, as well as classical (Goemans & Williamson, 1995; Rehfeldt et al., 2023; Benlic & Hao, 2013; Martí et al., 2009; Festa et al., 2002; Burer et al., 2002) and learned (Barrett et al., 2020; 2022; Zhang et al., 2023) algorithms. These efforts underscore the combination of intractability and broad applicability that motivates our focus on this problem.

To address these gaps and promote fair and comprehensive evaluation for CO, we carefully select distributions of instances for MaxCut and evaluate the performance of several highly cited and recently proposed heuristics on these instances. We pose the following research questions:

*In this setting, can we reproduce or corroborate the performance of learned heuristics as compared to their traditional counterparts? Is there any absolute performance gain, perhaps with a trade-off in computational efficiency, when using learned heuristics compared to reasonably effective traditional baselines? Finally, how well do algorithms trained with one distribution generalize to another distribution?*

**Contributions.**

- We provide an open-source benchmark suite **MaxCut-Bench** for the Maximum Cut problem. The software currently supports several highly cited or recently proposed, learned heuristics such as S2V-DQN (Khalil et al., 2017), ECO-DQN (Barrett et al., 2020), ANYCSP (Tönshoff et al., 2022), GFLOW-COMBOPT (Zhang et al., 2023). Our modern re-implementation of several algorithms, using efficient graph learning packages, is designed with a focus on reproducibility and scalability, enabling the handling of larger instances. Our evaluation suite is designed to support research on combinatorial problems on graphs and aims to provide a fair and comparable environment for future evaluations.
- Using MaxCut-Bench, we compare the learned heuristics to classical heuristics across multiple datasets that have previously been used to benchmark SOTA traditional heuristics. Our results reveal that classical local search heuristics, TABU SEARCH and EXTREMAL OPTIMIZATION, consistently match or outperform all evaluated learned heuristics in terms of objective value, scalability, and generalization—except for ANYCSP. This finding is particularly surprising given that S2V-DQN, for instance, is widely regarded as a strong heuristic, frequently used as a baseline, and cited thousands of times.
- Using our re-implementation of learned local search heuristics, we analyze deep learning techniques designed to enhance local search and conduct an exhaustive evaluation of these heuristics on hard instances. Our results demonstrate that the strong performance of the highly cited heuristic ECO-DQN can be replicated by selecting a subset of its features related to TABU SEARCH and replacing the GNN with a linear regression model, whether using the original code or our re-implementation.

The main goal of our work is to provide a fair and comparable environment for evaluating machine learning algorithms in combinatorial optimization, specifically for the Maximum Cut problem. We believe our work highlights the lack of standardized datasets and baselines necessary to demonstrate significant improvements. We hope that MaxCutBench can address this gap. In addition to our negative results, we also find that classic algorithms, such as local search, can be significantly enhanced with deep learning (Khalil et al., 2017; Yao et al., 2021). This suggests that more research is needed on replacing handcrafted rules with machine learning and standardization in datasets and baselines.

---

[1]This requires an exact solution to the MaxCut problem, and the 2021 Nobel Prize in Physics was awarded for work on spin glasses.

**Organization.** The rest of this paper is organized as follows. In Section 2, we discuss relevant related work. We present the MaxCut-Bench instance distributions and algorithms in Section 3. In Section 4, we use the benchmark to answer the motivating questions above. Finally, in Section 5, we conclude the paper.

## 2 Related Work

In the following, we review two categories of related works: benchmarking learned heuristics for CO, which aligns with our work, and algorithms for MaxCut.

**Benchmarking Learned Heuristics for CO.** With the growing interest in using GNNs to tackle CO problems, several recent works revisit their effectiveness compared to traditional heuristics. For the MaxCut problem, Yao et al. (2019) show that a simple local search algorithm, EXTREMAL OPTIMIZATION (Boettcher & Percus, 2001), consistently outperforms their adapted GNN baseline (Chen et al., 2017) across various configurations of dense and sparse random regular graphs. However, their study does not compare these results with existing solvers from the literature. To the best of our knowledge, we are the first to comprehensively compare SOTA-learned heuristics from the literature against classical heuristics for MaxCut.

For the MAXIMUM INDEPENDENT SET (MIS) problem, Böther et al. (2022) show that the performance of the popular guided tree search algorithm (Li et al., 2018) is not reproducible, and the GNN used in the tree search does not play any meaningful role. Instead, the various classical algorithms are responsible for the good performance, especially on hard instances. This is analogous to our result for the ECO-DQN algorithm (details in section 4.2), where we show that the GNN does not appear to play a meaningful role in the algorithm, and instead, the good performance of ECO-DQN for MaxCut may come from its Tabu Search-related features. Similarly, Angelini & Ricci-Tersenghi (2023) show that a simple greedy algorithm, running almost in linear time, finds solutions of better quality than a physics-inspired unsupervised GNN-based heuristic (Schuetz et al., 2022). In the context of the TRAVELING SALESMAN PROBLEM (TSP), Joshi et al. (2020) note that GNNs achieve promising results for relatively small instances, typically up to a few hundred cities. However, for instances involving millions of cities, the classical Lin-Kernighan-Helsgaun algorithm (Helsgaun, 2000; Taillard & Helsgaun, 2019) consistently finds solutions close to optimal. More recently, Liu et al. (2023) demonstrate that learned heuristics still lag behind traditional solvers in effectively solving TSP. For the MINIMUM VERTEX COVER problem, Sato et al. (2019) shows that the best approximation ratio achievable by a large class of GNNs is 2, which is suboptimal (Karakostas, 2009).

**Other Algorithms for MaxCut.** In addition to the algorithms included in this paper, several other algorithms are available for solving MaxCut. Dunning et al. (2018) systematically evaluate ten classical heuristics for MaxCut on a library of large instances and use machine learning to predict which heuristic works best on a previously unseen problem instance. Abe et al. (2019) combine a GNN with a Monte Carlo tree search approach to learn a heuristic for MaxCut. Beloborodov et al. (2020) use reinforcement learning in conjunction with a quantum-inspired algorithm to solve the Ising energy minimization problem, which is equivalent to the MaxCut problem. Karalias & Loukas (2020) propose an unsupervised approach with theoretical guarantees to produce a distribution over subsets of nodes as a possible solution to MaxCut. To address scalability challenges, Barrett et al. (2022) limits the costly GNN to a pre-processing step and introduces a recurrent unit for fast action selection. Schuetz et al. (2022) reformulate MaxCut as a quadratic unconstrained binary optimization problem, removing binary constraints to create a differentiable loss function. Readers may refer to Mazyavkina et al. (2021); Cappart et al. (2023) for further insights into learned heuristics for MaxCut.

## 3 The MaxCut-Bench Benchmark

In this section, we formally introduce the Maximum Cut (MaxCut) problem as follows. Given an undirected graph $G(V, E, w)$, where $V$ represents the set of vertices, $E$ denotes the set of edges and weights $w(u, v)$ on the edges $(u, v) \in E$, the goal is to find a subset of nodes $S \subseteq V$ that maximizes the objective function, $f(S) = \sum_{u \in S, v \in V \setminus S} w(u, v)$. Next, we present a comprehensive overview of the setup for **MaxCut-Bench**. We provide the general selection criteria and considerations for algorithms (Section 3.1) and datasets (Section 3.2).

Table 1: Categorization of all algorithms and datasets used in our evaluation.

| Algorithms | |
|---|---|
| IP solver | CPLEX, GUROBI |
| Classical Heuristics | SEMIDEFINITE PROGRAMMING, GREEDY, TABU SEARCH, EXTREMAL OPTIMIZATION |
| GNNs | S2V-DQN, ECO-DQN, LS-DQN, RUN-CSP, ANYCSP, GFLOW-COMBOPT |
| Quantum Annealing | Leleu et al. (2019), Leleu et al. (2021) |
| **Datasets** | |
| Random Instaces | Erdős-Rényi, Barabási-Albert, Holme-Kim, Watts-Strogatz |
| Hard Instances | Gset, SK spin glass, Optsicom, Phase transition, Big Mac Library, MaxCut and BQP Instance Library |

### 3.1 Benchmark Algorithms

Table 1 shows the overall 16 algorithms integrated into our benchmark. They are divided into four categories and we briefly introduce each category in the following, and more details are provided in Appendix A.2.

**Classical methods.** We include common classical algorithms, such as SEMIDEFINITE PROGRAMMING (SDP) and GREEDY, along with two local search algorithms: TABU SEARCH (TS) (Glover, 1990) and EXTREMAL OPTIMIZATION (EO) (Boettcher & Percus, 2001). Both TS and EO incorporate strategies to escape local minima and explore the search space more effectively.

**Mixed-Integer program solver.** We select two commonly used commercial mixed-integer programming solvers: CPLEX and GUROBI.

**GNN-based heuristics.** We focus on works selected based on three primary criteria: being highly cited, recently introduced, or widely used as benchmarks for evaluating learned heuristics. S2V-DQN, introduced by Khalil et al. (2017), is a seminal algorithm with thousands of citations and serves as a baseline in numerous subsequent studies (Li et al., 2018; Manchanda et al., 2019; Barrett et al., 2020; Sun et al., 2023). In addition to replicating S2V-DQN, we include its extensions, ECO-DQN (Barrett et al., 2020) and LS-DQN (Yao et al., 2021). Furthermore, we incorporate RUN-CSP (Toenshoff et al., 2021), a common baseline for recent neural solvers (Schuetz et al., 2022; Sun et al., 2022; Toenshoff et al., 2021). Finally, we include two recent works: GFLOW-COMBOPT (Zhang et al., 2023), highlighted at NeurIPS 2023, and ANYCSP (Tönshoff et al., 2022), published in IJCAI 2023.

**Quantum annealing.** We use two SOTA quantum annealing algorithms (Leleu et al., 2019; 2021) to obtain the best (possibly optimal) solutions within a cutoff time of 1 hour. All approximation ratios reported in the paper are based on the best solution found by either of the two solvers.

### 3.2 Benchmark Datasets

We have gathered a diverse collection of instances from both real-world datasets and random graph distributions, as detailed in Table 1, which are based on the following criteria:

**Random Instances.** To ensure a thorough comparison, we use commonly employed random graph models for instance generation, including Erdős et al. (1960) (ER), Albert & Barabási (2002) (BA), Holme & Kim (2002) (HK) and Watts & Strogatz (1998) (WS).

**Hard Instances.** We select datasets that have been employed for evaluating the performance of several SOTA traditional heuristics (Benlic & Hao, 2013; Leleu et al., 2019; 2021; Hamerly et al., 2019). These datasets include Sherrington-Kirkpatrick (SK) spin glass (Sherrington & Kirkpatrick, 1975), dense unweighted instances at their phase transitions (Coppersmith et al., 2004), Optsicom project (Corberán et al., 2006), Big Mac Library (Rendl et al., 2010), MaxCut and BQP Instance Library (Bonn, 2010) and Stanford Gset dataset (Ye, 2003).

For a more comprehensive description of all mentioned datasets and the hyperparameters used for graph generation, please refer to Appendix A.1.

## 4 Experiments and Analysis

In this section, we use the benchmark to investigate the **objective value**, **inference time**, **scalability** and **generalization** of algorithms by answering the following specific questions:

- *How effective are the algorithms frequently used as baselines in the literature?* We observe that commonly used baselines for evaluating learned heuristics, such as CPLEX, SDP, and GREEDY, often fail to provide strong comparisons, as their performance is frequently surpassed by simple stochastic local search algorithms, such as TS and EO (Section 4.1).
- *Does deep learning combined with classical heuristics improve the performance of classical heuristics?* While S2V-DQN and LS-DQN demonstrate improvements over local search by incorporating GNNs, we have noted that the GNN in ECO-DQN can be effectively replaced by linear regression without any loss in performance. (Section 4.2).
- *Can learned heuristics consistently outperform simple, general-purpose heuristics?* Except for ANYCSP, the answer is generally no. We demonstrate that several learned heuristics do not outperform TS and EO (Section 4.3).
- *Can learned heuristics maintain their effectiveness when applied to out-of-distribution instances beyond their training data?* Again, except for ANYCSP, the answer is no. We observe a substantial decline in performance when learned heuristics are tested on graph distributions different from the training set, with ANYCSP being a notable exception (Section 4.4).
- *How efficient are these algorithms in terms of time and space on large hard instances?* Classical local search heuristics tend to outperform most learned heuristics while using minimal computing resources and inference time, although their performance does decrease as the problem size increases (Section 4.5).

All learned algorithms in our benchmark are implemented using PyTorch (Paszke et al., 2019). All experiments were conducted on a Linux server with a GPU (NVIDIA RTX A6000) and CPU (AMD EPYC 7713), using PyTorch 2.3.0, DGL 2.2.1 (Wang, 2019) and Python 3.11.9. We clearly state the changes necessary to update previous implementations and ensure that our reimplementations align with the results published in the literature. We provide all the details of these changes in Appendix A.4.

**Evaluation settings.** All training is performed on randomly 4000 generated graphs and the validation is performed on a fixed set of 50 held-out graphs from the same distribution. For synthetic datasets, testing is performed on 100 instances drawn from the same distribution; or upon the test instances provided in the original resource (see details in Appendix A.1). We evaluate the algorithms using the average approximation ratio as a performance metric. Given an instance, the approximation ratio for an algorithm is computed by normalizing the objective value of the best-known solution for the instance. As S2V-DQN is deterministic at test time, we use a single optimization episode for each graph. For all other algorithms, we run each algorithm for 50 randomly initialized episodes with $2|V|$ number of search steps per episode and select the best outcome from these runs following the experimental setup of Barrett et al. (2020) and Yao et al. (2021). Experimentally, we have found that the performance of all algorithms saturates within this number of search steps (we refer the reader to Appendix A.6 for more details).

**Summary of results.** In Figure 1, we present an overview of the performance of all heuristics, excluding the two quantum annealing algorithms specifically used to find the best-known solutions with a one-hour cutoff time. Our key findings include the following: classical heuristics are often faster while providing solutions of similar or better quality; the GNN used in ECO-DQN can be replaced by linear regression without any performance degradation; and a recent solver, ANYCSP, based on global search, can provide competitive solution quality but requires significantly greater computational resources.

### 4.1 Effectiveness of Common Baselines in Evaluating Learned Heuristics

In this section, we analyze the performance of the common classical baselines that are used for comparing the performance of learned heuristics for MaxCut. The main question we address is whether outperforming certain heuristics is significant. For this evaluation, we use the dataset from Barrett et al. (2020), where

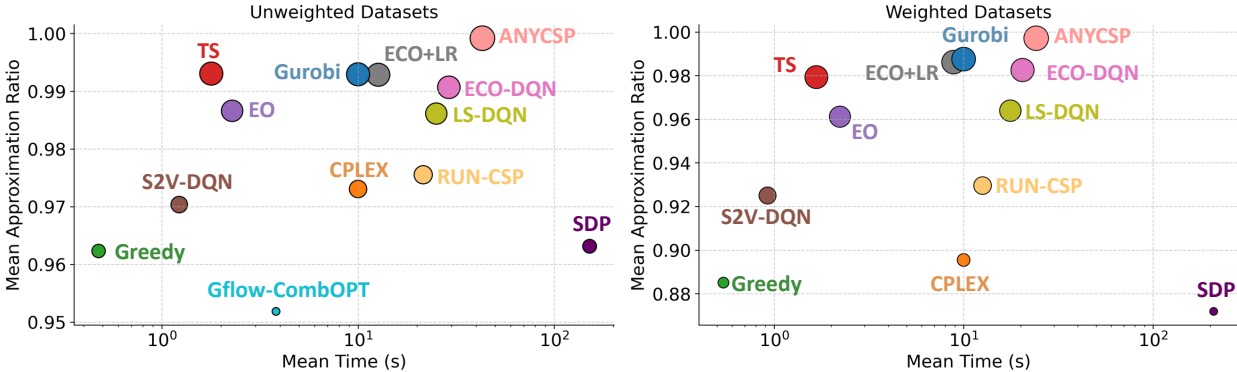

Figure 1: Comparison of the performance between learned and classical heuristics. Classical heuristics, such as EO and TS, often outperform most learned heuristics while being faster. Note that we include GREEDY, which starts with a random solution rather than an empty one, as opposed to Zhang et al. (2023).

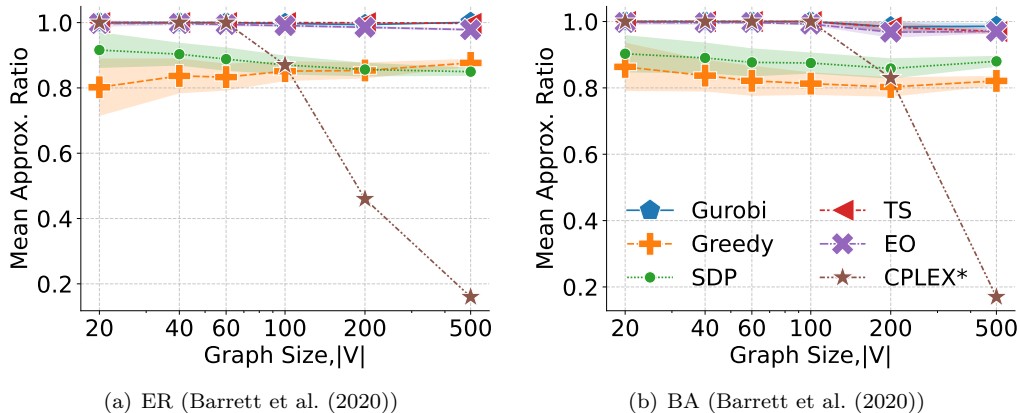

(a) ER (Barrett et al. (2020))     (b) BA (Barrett et al. (2020))

Figure 2: Performance comparison of classical baselines. The most common baselines, GREEDY, SDP, and CPLEX, can be easily outperformed by classical local search heuristics ; *Values (10-minute cut-off time) as reported in Barrett et al. (2020)

optimal or best-known solutions are publicly available. This dataset consists of ER and BA graphs up to 500 vertices.

As shown in Fig. 2, CPLEX performs relatively well on graphs with up to 60 vertices but shows a significant drop in performance for graphs with 200 and 500 vertices due to the large solution space. A similar drop in CPLEX performance for larger graphs was reported by Khalil et al. (2017). The GREEDY algorithm is robust across distributions but typically performs 20% worse than the best-known solutions. Similarly, SDP performs slightly better than GREEDY but still exhibits an optimality gap of 10–15%. In contrast, GUROBI finds solutions close to the best-known values very quickly, within a 10-second time limit.

EO and TS consistently outperform or match GREEDY, SDP and CPLEX, demonstrating their competitiveness as heuristics. However, their performance declines as instance size increases (see Fig. 3).

**Final Considerations.** We conclude that while improvements over commonly used baselines, such as GREEDY, SDP, and CPLEX, are incremental, simple modifications to local search can outperform these baselines by a significant margin. This highlights the need for more work to standardize baseline selection for the empirical evaluation of CO problems.

Table 2: Summary of the learned algorithms that improve upon local search.

| Algorithm | Initial Solution | Node Features | Action | Reward | Termination | Simplified Algorithm |
|-----------|------------------|---------------|--------|--------|-------------|----------------------|
| S2V-DQN | Empty | Vertex State | Add | Marginal gain | all marginal gains are negative | Greedily add, starting with empty solution |
| LS-DQN | Random | Vertex State | Add or Delete | -Marginal gain | Self-termination or $2|V|$ steps | Greedily add/delete, starting with random solution |
| ECO-DQN | Random | Vertex State Marginal Gain Time Since Flip .... | Add or Delete | max((current objective value-best objective value),0)+ small reward if new local minima is found | $2|V|$ steps | TABU SEARCH |

## 4.2 Impact of Integrating Deep Learning with Local Search

In this subsection, we examine whether deep learning heuristics that combine local search with a neural network provide any improvement over local search alone. The pipeline of this category of algorithms is as follows: handcrafted node features are generated at each step of the algorithm, and passed through a GNN to generate node embeddings. These embeddings are fed into a reinforcement learning (RL) algorithm, which selects the action to take. The action involves either selecting a node to add to or delete from the solution. Table 2 summarizes the key differences between these algorithms. Next, we discuss each algorithm and investigate the causes of some of the unexpected outcomes presented in our paper.

**S2V-DQN.** S2V-DQN starts with an empty solution and incrementally adds a vertex, guided by a GNN until no greedy actions remain. The concept can be simplified to greedily adding the node that results in the largest improvement in objective value; we refer to this as S2V-Simplified. From Fig. 3, we observe that S2V-DQN outperforms GREEDY and S2V-Simplified in four out of five distributions from the well-known Gset benchmark dataset, typically by a margin of 10-20%. Since GREEDY was reported as the best competitor to S2V-DQN by Khalil et al. (2017), this confirms that our implementation successfully reproduces the performance reported in the original work (See more details in A.4.1). This improvement is impressive, showing the promise of using GNNs to improve classical heuristics.

However, from Fig. 3 and Table 3, we observe that TS and EO —two simple local search heuristics with a single parameter— outperform S2V-DQN in almost all cases, typically by a margin of 5-10%. In fact, even starting from a random solution and applying the GREEDY algorithm outperforms S2V-DQN on certain distributions, usually by a small margin, such as ER (weighted and unweighted), BA (unweighted), and Phase Transition networks, as shown in Table 3. This demonstrates the power of stochasticity when combined with local search.

**LS-DQN.** Following in the footsteps of S2V-DQN, LS-DQN starts with a random solution and adds or deletes from a node from the solution at each step guided by a GNN. LS-DQN improves the performance of its simplified version as shown in Table 3, especially for weighted instances.

However, our analysis contradicts the claim of Yao et al. (2021) that LS-DQN matches the performance of ECO-DQN. This may be explained by observing that Yao et al. (2021) compared the performance of ECO-DQN and LS-DQN only on complete graphs, which underscores the importance of comparing algorithms across a range of instances.

**ECO-DQN.** While S2V-DQN and LS-DQN use the current solution as the state-space representation for the RL agent, ECO-DQN proposes six additional handcrafted features per vertex for its state space. We observe that two of these features are closely related to the traditional heuristic, TABU SEARCH (TS): **1) Marginal Gain:** the change in the objective value when a vertex is added to or removed from the solution set (we refer this action as a flip) **2) Time Since Flip:** steps since the vertex has last been flipped to prevent short looping trajectories.

We first describe how the TS algorithm works to clarify why TS can be considered a simpler version of ECO-DQN. TS selects the node to flip that results in the maximum increase in the objective value, provided the time since the last flip of the node exceeds a specified hyperparameter. This approach helps TS actively

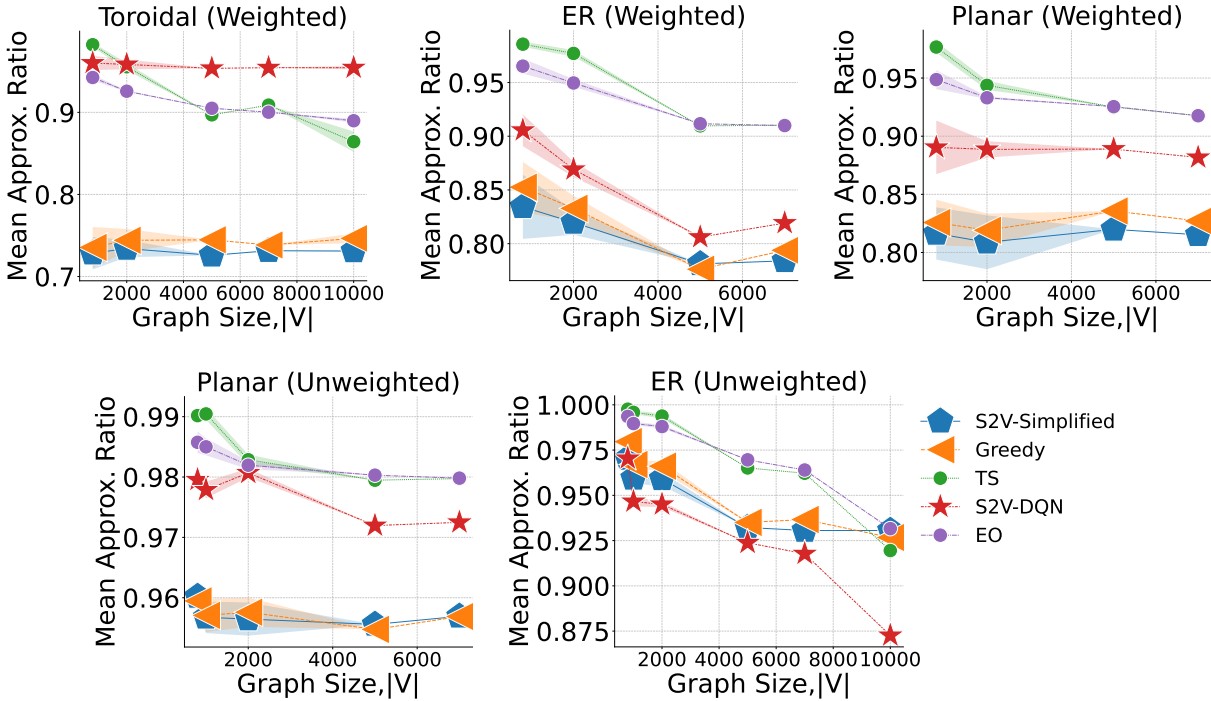

Figure 3: Performance comparison of S2V-DQN and classical approaches. S2V-DQN outperforms its simplified version and GREEDY by a significant margin, consistent with Khalil et al. (2017). See Table 11.

Table 3: Performance comparison of learned heuristics with their simple counterparts (best in **bold**): The first and second halves of the table show results for unweighted and weighted instances, respectively.

| | | S2V-DQN | | LS-DQN | | ECO-DQN | | |
|---|---|---|---|---|---|---|---|---|
| Graph | Nodes | S2V-Simplified | S2V+GNN | GREEDY | LS+GNN | TS | ECO+LR | ECO+GNN |
| Gset (ER) | 800 | $0.970^{\pm0.003}$ | $0.970^{\pm0.001}$ | $0.985^{\pm0.001}$ | $0.992^{\pm0.001}$ | $\mathbf{0.999^{\pm0.001}}$ | $0.996^{\pm0.001}$ | $0.997^{\pm0.001}$ |
| Gset (Planar) | 800 | $0.960^{\pm0.002}$ | $0.980^{\pm0.001}$ | $0.963^{\pm0.002}$ | $0.984^{\pm0.002}$ | $\mathbf{0.992^{\pm0.001}}$ | $0.991^{\pm0.001}$ | $0.990^{\pm0.001}$ |
| BA | 800 | $0.938^{\pm0.007}$ | $0.951^{\pm0.006}$ | $0.952^{\pm0.003}$ | $0.983^{\pm0.003}$ | $0.992^{\pm0.002}$ | $\mathbf{0.993^{\pm0.002}}$ | $0.991^{\pm0.002}$ |
| WS | 800 | $0.944^{\pm0.006}$ | $0.973^{\pm0.005}$ | $0.927^{\pm0.003}$ | $0.971^{\pm0.003}$ | $0.990^{\pm0.002}$ | $0.988^{\pm0.002}$ | $\mathbf{0.992^{\pm0.002}}$ |
| HK | 800 | $0.939^{\pm0.007}$ | $0.966^{\pm0.005}$ | $0.951^{\pm0.003}$ | $0.984^{\pm0.003}$ | $\mathbf{0.992^{\pm0.002}}$ | $\mathbf{0.992^{\pm0.002}}$ | $\mathbf{0.992^{\pm0.002}}$ |
| Phase Transition | 100-200 | $0.982^{\pm0.005}$ | $0.985^{\pm0.006}$ | $0.996^{\pm0.002}$ | $0.998^{\pm0.002}$ | $\mathbf{1.000^{\pm0.000}}$ | $\mathbf{1.000^{\pm0.000}}$ | $\mathbf{1.000^{\pm0.000}}$ |
| Gset (ER) | 800 | $0.856^{\pm0.026}$ | $0.906^{\pm0.017}$ | $0.911^{\pm0.012}$ | $0.953^{\pm0.007}$ | $\mathbf{0.995^{\pm0.003}}$ | $0.981^{\pm0.002}$ | $0.984^{\pm0.003}$ |
| Gset (Planar) | 800 | $0.863^{\pm0.030}$ | $0.890^{\pm0.027}$ | $0.871^{\pm0.013}$ | $0.942^{\pm0.007}$ | $\mathbf{0.980^{\pm0.005}}$ | $0.975^{\pm0.005}$ | $0.966^{\pm0.009}$ |
| Gset (Toroidal) | 800 | $0.838^{\pm0.003}$ | $0.960^{\pm0.010}$ | $0.793^{\pm0.010}$ | $0.965^{\pm0.007}$ | $0.989^{\pm0.003}$ | $0.993^{\pm0.004}$ | $\mathbf{0.994^{\pm0.002}}$ |
| BA | 800 | $0.853^{\pm0.018}$ | $0.886^{\pm0.019}$ | $0.855^{\pm0.012}$ | $0.934^{\pm0.011}$ | $0.978^{\pm0.007}$ | $\mathbf{0.979^{\pm0.008}}$ | $0.972^{\pm0.009}$ |
| WS | 800 | $0.861^{\pm0.014}$ | $0.919^{\pm0.012}$ | $0.833^{\pm0.010}$ | $0.923^{\pm0.007}$ | $0.967^{\pm0.006}$ | $\mathbf{0.973^{\pm0.006}}$ | $0.961^{\pm0.008}$ |
| HK | 800 | $0.857^{\pm0.019}$ | $0.908^{\pm0.017}$ | $0.855^{\pm0.012}$ | $0.937^{\pm0.009}$ | $0.977^{\pm0.007}$ | $\mathbf{0.978^{\pm0.008}}$ | $0.974^{\pm0.009}$ |
| Barrett et al. (ER) | 200 | $0.866^{\pm0.038}$ | $0.951^{\pm0.024}$ | $0.954^{\pm0.014}$ | $0.987^{\pm0.010}$ | $\mathbf{1.000^{\pm0.001}}$ | $\mathbf{1.000^{\pm0.001}}$ | $\mathbf{1.000^{\pm0.001}}$ |
| Barrett et al. (BA) | 200 | $0.849^{\pm0.054}$ | $0.937^{\pm0.043}$ | $0.903^{\pm0.039}$ | $0.977^{\pm0.032}$ | $\mathbf{0.984^{\pm0.032}}$ | $\mathbf{0.984^{\pm0.032}}$ | $0.983^{\pm0.033}$ |
| SK spin-glass | 70-100 | $0.865^{\pm0.057}$ | $0.939^{\pm0.049}$ | $0.994^{\pm0.010}$ | $0.999^{\pm0.003}$ | $\mathbf{1.000^{\pm0.000}}$ | $\mathbf{1.000^{\pm0.000}}$ | $\mathbf{1.000^{\pm0.000}}$ |
| Optsicom | 125 | $0.779^{\pm0.049}$ | $0.962^{\pm0.023}$ | $0.872^{\pm0.022}$ | $0.991^{\pm0.010}$ | $\mathbf{1.000^{\pm0.000}}$ | $\mathbf{1.000^{\pm0.000}}$ | $\mathbf{1.000^{\pm0.000}}$ |

avoid revisiting previously explored points in the search space. However, if flipping a node results in the best objective value found so far, TS overrides this restriction and flips that node.

Now looking at the reward function defined in ECO-DQN, we see the algorithm trains its agent to avoid revisiting the same local minima by providing a small reward (referred to as the immediate reward in the original paper) when the agent discovers a new local minimum. Additionally, ECO-DQN encourages the agent to select the node that gives the best objective value found so far. Intuitively, this reward mechanism

should encourage the selection of nodes with a high time since the last flip and those that yield the best objective value.

To better understand the performance of ECO-DQN, we compare it with TS and an ablated version of ECO-DQN, called SoftTabu (ECO+LR). SoftTabu replaces the GNN with linear regression and omits the features not related to TS, that is, all but the two features described above.

From Table 3, we observe that TS and SoftTabu often match or even outperform ECO-DQN (ECO+GNN) in terms of performance. To further validate this observation, we conduct empirical experiments based on the original paper, using the publicly available implementation of ECO-DQN provided by the authors without any modifications. The results are then compared with TS and ECO+LR (see Appendix A.4.2).

We use a paired t-test to compare different methods and conclude that there is no significant performance difference between ECO-DQN and its simpler counterparts (as shown in Table 12). In fact, adding the GNN adversely affects the generalization performance of ECO-DQN (as discussed in Section 4.4).

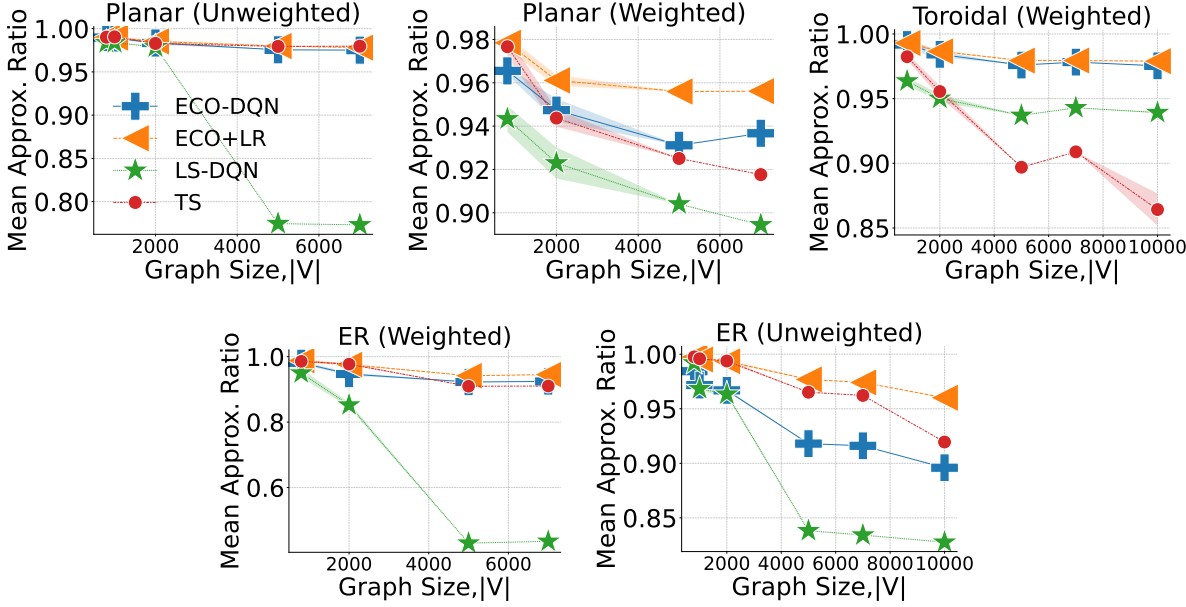

Figure 4: Performance comparison of ECO-DQN, LS-DQN, and classical approaches. Replacing GNN in ECO-DQN with linear regression does not hurt performance, underscoring the need for ablation studies. Results are also shown in Table 11.

**Ablation Studies in Algorithm Design.** We observe that the highly cited ECO-DQN does not exhibit performance gains from the inclusion of deep learning. To clearly establish the contribution of the deep learning component, we strongly recommend conducting ablation studies for any proposed algorithm.

**A New Approach to Heuristic Optimization.** S2V-DQN and its subsequent refinement, LS-DQN, serve as proof of concept that, despite not surpassing all baselines, they introduce a novel paradigm for improving heuristics by replacing handcrafted rules with machine learning. At the same time, our extensive analysis using MaxCutBench underscores the importance of benchmarking against simple heuristics to gain meaningful insights and measure progress comprehensively.

**Final Considerations.** As instance sizes increase, the performance gap between the best-known solutions across learned heuristics grows significantly. This limitation highlights the challenge of enabling agents to reason effectively about larger systems, creating opportunities for future research.

Table 4: Average Approximation ratios of classical and learned heuristics (best in bold);"—" denotes no reasonable result is achieved by the corresponding algorithm and the first and second halves of the table show results for unweighted and weighted instances, respectively.

| Graph | Nodes | Classical Heuristics | | Learned Heuristics | | | | | |
|---|---|---|---|---|---|---|---|---|---|
| | | EO | TS | S2V-DQN | LS-DQN | ECO-DQN | RUN-CSP | GFLOW-COMBOPT | ANYCSP |
| Gset (ER) | 800 | $0.995^{\pm 0.001}$ | $\mathbf{0.999^{\pm 0.001}}$ | $0.970^{\pm 0.001}$ | $0.970^{\pm 0.001}$ | $0.997^{\pm 0.001}$ | $0.982^{\pm 0.001}$ | $0.979^{\pm 0.003}$ | $\mathbf{0.999^{\pm 0.000}}$ |
| Gset (Planar) | 800 | $0.987^{\pm 0.000}$ | $0.992^{\pm 0.001}$ | $0.980^{\pm 0.001}$ | $0.980^{\pm 0.001}$ | $0.990^{\pm 0.001}$ | $0.958^{\pm 0.003}$ | $0.967^{\pm 0.003}$ | $\mathbf{0.997^{\pm 0.001}}$ |
| BA | 800 | $0.983^{\pm 0.003}$ | $0.992^{\pm 0.002}$ | $0.951^{\pm 0.006}$ | $0.951^{\pm 0.006}$ | $0.991^{\pm 0.002}$ | $0.978^{\pm 0.003}$ | $0.955^{\pm 0.006}$ | $\mathbf{1.000^{\pm 0.000}}$ |
| WS | 800 | $0.979^{\pm 0.003}$ | $0.990^{\pm 0.002}$ | $0.973^{\pm 0.005}$ | $0.973^{\pm 0.005}$ | $0.992^{\pm 0.002}$ | $0.978^{\pm 0.003}$ | $0.958^{\pm 0.004}$ | $\mathbf{0.999^{\pm 0.000}}$ |
| HK | 800 | $0.983^{\pm 0.003}$ | $0.992^{\pm 0.002}$ | $0.966^{\pm 0.005}$ | $0.966^{\pm 0.005}$ | $0.992^{\pm 0.002}$ | $0.977^{\pm 0.003}$ | $0.930^{\pm 0.015}$ | $\mathbf{1.000^{\pm 0.000}}$ |
| Phase Transition | 100-200 | $0.999^{\pm 0.001}$ | $\mathbf{1.000^{\pm 0.000}}$ | $0.985^{\pm 0.006}$ | $0.985^{\pm 0.006}$ | $\mathbf{1.000^{\pm 0.000}}$ | $0.994^{\pm 0.002}$ | $0.993^{\pm 0.004}$ | $\mathbf{1.000^{\pm 0.000}}$ |
| Gset (ER) | 800 | $0.973^{\pm 0.005}$ | $0.995^{\pm 0.003}$ | $0.906^{\pm 0.017}$ | $0.906^{\pm 0.017}$ | $0.984^{\pm 0.003}$ | $0.909^{\pm 0.007}$ | — | $\mathbf{0.998^{\pm 0.002}}$ |
| Gset (Planar) | 800 | $0.954^{\pm 0.007}$ | $0.980^{\pm 0.005}$ | $0.890^{\pm 0.027}$ | $0.890^{\pm 0.027}$ | $0.966^{\pm 0.009}$ | $0.925^{\pm 0.013}$ | — | $\mathbf{0.995^{\pm 0.005}}$ |
| Gset (Toroidal) | 800 | $0.948^{\pm 0.010}$ | $0.989^{\pm 0.003}$ | $0.960^{\pm 0.010}$ | $0.960^{\pm 0.010}$ | $0.994^{\pm 0.002}$ | $0.976^{\pm 0.002}$ | — | $\mathbf{0.999^{\pm 0.002}}$ |
| BA | 800 | $0.949^{\pm 0.008}$ | $0.978^{\pm 0.007}$ | $0.886^{\pm 0.019}$ | $0.886^{\pm 0.019}$ | $0.972^{\pm 0.009}$ | $0.931^{\pm 0.009}$ | — | $\mathbf{1.000^{\pm 0.000}}$ |
| WS | 800 | $0.943^{\pm 0.007}$ | $0.967^{\pm 0.006}$ | $0.919^{\pm 0.012}$ | $0.919^{\pm 0.012}$ | $0.961^{\pm 0.008}$ | $0.950^{\pm 0.007}$ | — | $\mathbf{0.999^{\pm 0.000}}$ |
| HK | 800 | $0.949^{\pm 0.009}$ | $0.977^{\pm 0.007}$ | $0.908^{\pm 0.017}$ | $0.908^{\pm 0.017}$ | $0.974^{\pm 0.009}$ | $0.937^{\pm 0.009}$ | — | $\mathbf{1.000^{\pm 0.000}}$ |
| Barrett et al. (ER) | 200 | $0.989^{\pm 0.008}$ | $\mathbf{1.000^{\pm 0.000}}$ | $0.951^{\pm 0.024}$ | $0.951^{\pm 0.024}$ | $\mathbf{1.000^{\pm 0.001}}$ | $0.940^{\pm 0.015}$ | — | $\mathbf{1.000^{\pm 0.000}}$ |
| Barrett et al. (BA) | 200 | $0.969^{\pm 0.035}$ | $\mathbf{1.000^{\pm 0.032}}$ | $0.937^{\pm 0.043}$ | $0.937^{\pm 0.043}$ | $0.983^{\pm 0.033}$ | $0.958^{\pm 0.035}$ | — | $0.986^{\pm 0.032}$ |
| SK spin-glass | 70-100 | $0.995^{\pm 0.006}$ | $\mathbf{1.000^{\pm 0.000}}$ | $0.939^{\pm 0.049}$ | $0.939^{\pm 0.049}$ | $\mathbf{1.000^{\pm 0.000}}$ | $0.962^{\pm 0.019}$ | — | $\mathbf{1.000^{\pm 0.001}}$ |
| Physics (Regular) | 125 | $0.986^{\pm 0.011}$ | $\mathbf{1.000^{\pm 0.000}}$ | $0.962^{\pm 0.023}$ | $0.962^{\pm 0.023}$ | $\mathbf{1.000^{\pm 0.000}}$ | $0.989^{\pm 0.009}$ | — | $\mathbf{1.000^{\pm 0.000}}$ |

## 4.3 Performance Comparison of Learned Heuristics and Simple Heuristics

In this section, we evaluate the algorithms to assess whether any of the deep learning heuristics can achieve the SOTA objective value on the instances in the MaxCut benchmark. While it may be too early to expect learned heuristics to beat SOTA heuristics tailored for specific problems, the question of how learned heuristics fare against simple heuristics remains unanswered. One of the main reasons for this is the variety of baselines and instances used in existing work (more details can be found in Appendix A.3).

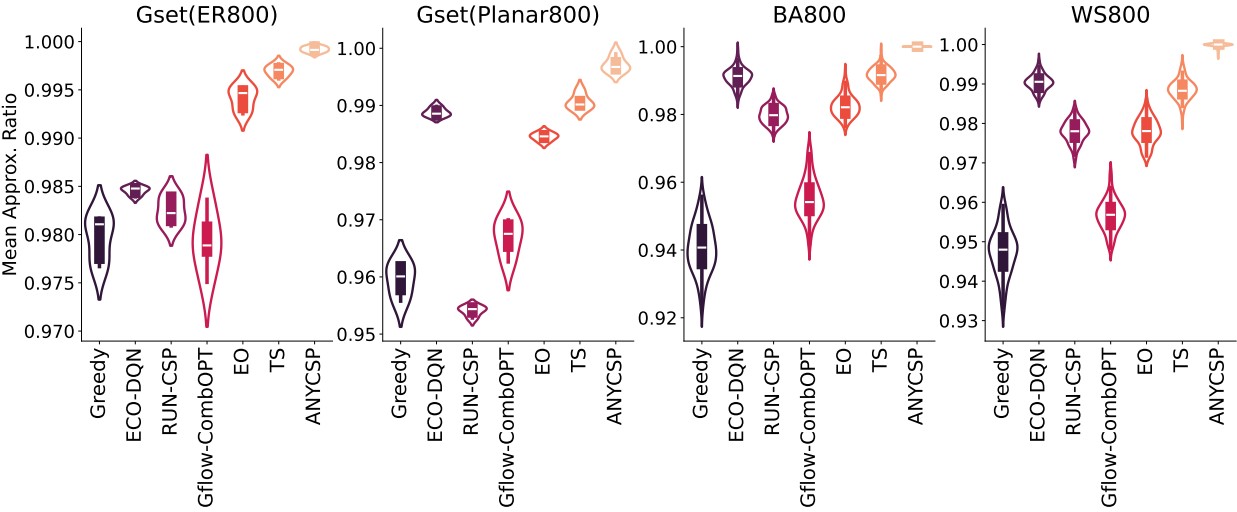

Figure 5: Violin plots of mean approx. ratio for learned and classical heuristics on a selection of unweighted instances. Among the learned heuristics, only ANYCSP outperforms the classical heuristics.

**Impact of findings.** We compare learned local search heuristics (S2V-DQN, ECO-DQN and LS-DQN), GFLOW-COMBOPT, RUN-CSP, and ANYCSP with the traditional heuristics EO and TS. Our results show that EO and TS can either outperform or achieve comparable solution quality to learned local search heuristics. Additionally, we observe that EO and TS can outperform GFLOW-COMBOPT and RUN-CSP, especially on larger instances, while ANYCSP usually outperforms them by a margin of 2-5%. Next, we discuss our empirical observations for each algorithm.

**Learned Local Search Heuristics.** Similar to our observation for TS in Section 4.2, EO can match or even surpass the performance of learned local search heuristics. While ECO-DQN performs slightly better

than EO, this outcome is expected, as TS also outperforms EO, and no significant differences are observed between these methods.

**RUN-CSP.** From Table 4, we observe that TS outperforms RUN-CSP on all distributions, while EO outperforms RUN-CSP on most distributions. Although RUN-CSP is commonly used as a baseline in recent neural CO solvers (Karalias & Loukas, 2020; Sun et al., 2022; Schuetz et al., 2022; Tönshoff et al., 2022), it is worth noting that its performance can be readily exceeded by classical heuristics. This underscores the importance of choosing robust baselines to highlight meaningful improvements.

**Gflow-CombOpt.** We restrict our empirical evaluation of GFLOW-COMBOPT to unweighted instances because Zhang et al. (2023) only evaluated GFLOW-COMBOPT for MaxCut on unweighted BA graphs, following Sun et al. (2022), and our attempt to adapt GFLOW-COMBOPT to handle weighted graphs performed poorly and did not surpass GREEDY. From Table 4 and Figure 5, we observe that TS and EO consistently outperform GFLOW-COMBOPT. In our opinion, this example highlights the need for more standardization in evaluation, as GFLOW-COMBOPT is often outperformed by naive local search heuristics.

**ANYCSP.** We analyze ANYCSP in its default configuration and observe that it consistently finds near-optimal solutions (that is, solutions near the value found by the Quantum Annealing algorithms), and is superior in value to TS and the other traditional local search algorithms. These promising results demonstrate that ANYCSP can effectively learn the solution structure across diverse graph distributions and remains robust across multiple distributions.

**Final considerations.** The empirical findings suggest that benchmarking against weak heuristics on a very particular set of instances may establish a low standard, potentially leading to a misleading sense of achievement. Hence, selecting appropriate baselines is crucial to accurately assess the effectiveness of learned heuristics. At a minimum, we recommend that any work should at least include simple algorithms that improve upon GREEDY– failure to consistently beat the naive baselines suggests that the algorithm may need further development.

## 4.4 Generalization Capabilities of Learned Heuristics

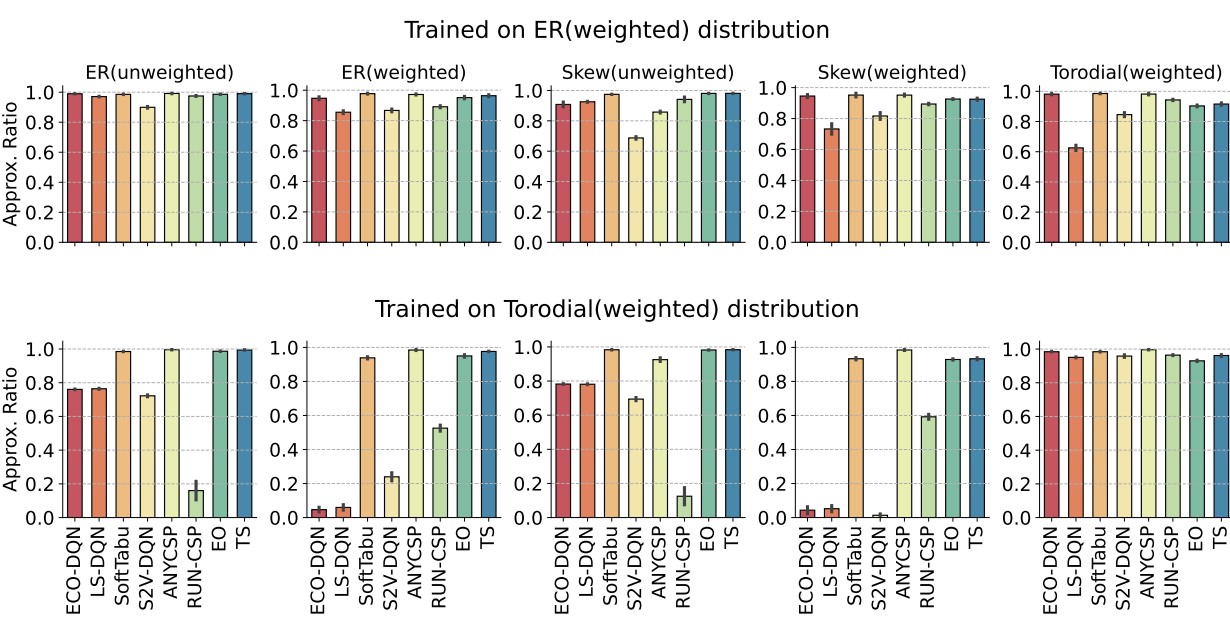

Figure 6: Generalization of algorithms for unseen distribution. The inclusion of GNN in ECO-DQN hurts generalization, and except for ANYCSP, all other heuristics perform poorly.

The ability of learned heuristics to perform well on a wide range of distributions, even if these distributions are not represented during training, is a highly desirable characteristic for practical CO problems (Cappart

et al., 2023). Many works include experiments to assess how well the heuristic generalizes. Often, this is in the form of training on smaller instances and generalizing to test on larger ones, although cross-distribution performance is also frequently assessed.

Several learned heuristics such as ECO-DQN, LS-DQN, and S2V-DQN are claimed to exhibit promising performance across a diverse range of graph structures, including those not present in their training data. In this section, we evaluate the generalization performance using **MaxCut-Bench**.

**Impact of Findings.** From Figure 6, we notice that there can be a substantial decline in performance when the learned heuristics are tested on graph distributions other than train distributions, with the notable exception of ANYCSP. In particular, observe that when trained on the toroidal distribution (second row), the test performance of ECO-DQN, LS-DQN, S2V-DQN, and RUN-CSP may fall below 25% on several distributions. This outcome may be anticipated. Intuitively, we would expect a network trained on instances of a particular structure to adapt toward this class of instances and perform poorer for different structures. We observe that TS, SoftTabu, and EO seem to generalize well across wider distributions. Both TS and EO have a single parameter, which we optimized for the training distribution to assess its generalization performance.

These results raise the possibility that the generalization of learned heuristics from learning over small and easy instances to testing on larger and more complicated ones may not be as robust as the literature (Khalil et al., 2017; Barrett et al., 2020; 2022; Yao et al., 2021) suggests. This feature is often touted as an amelioration of the expensive training process required for the learned heuristics. Additional results from our generalization experiments on BigMac, MaxCut, and the BQP Instance Library can be found in A.5.

### 4.5 Efficiency and Trade-offs in Large-Scale Instances

In this section, we analyze the efficiency and scalability of learned heuristics over ER graphs of size $|V| = 800$ from the Gset dataset. For time efficiency, we evaluate the efficiency of the algorithms by measuring the wall-clock time. For scalability, we evaluate the average GPU and CPU usage per second of these learned algorithms. Due to space constraints, we report run-time and performance of each algorithm across datasets in Appendix 10.

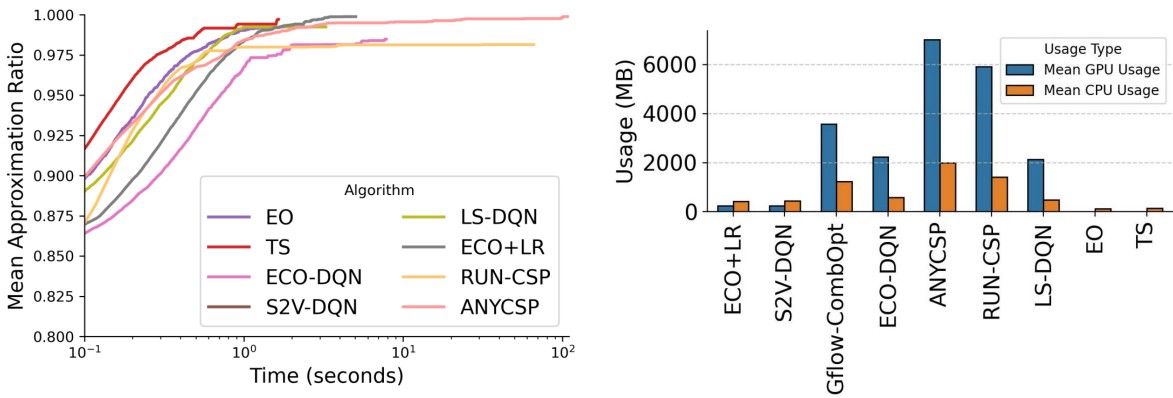

Figure 7: Comparison of the wall-clock time and average GPU and CPU memory utilization among heuristics. Iterative approaches, such as ANYCSP, ECO-DQN, and LS-DQN, take significantly more time.

**Impact of Findings.** From Figure 7, we observe that algorithms that use multiple randomly initialized episodes (iterative approaches) require significantly more time and memory. We observe that ANYCSP takes the longest to complete and consumes the most memory, while classical heuristics are an order of magnitude faster. This demonstrates a clear trade-off where the more accurate algorithms come with a longer run-time.

## 5  Conclusion

In this paper, we introduce **MaxCut-Bench**, a comprehensive benchmark for evaluating deep learning-based algorithms for the Maximum Cut problem, consisting of carefully selected instance distributions and

implemented algorithms. We intend that **MaxCut-Bench** will foster further research and refinement of learning-based algorithms, enabling more informed evaluations and comparisons. We regard our work as a long-term evolving project and are dedicated to its continuous development. Our roadmap for the future includes expanding its scope to cover a broader spectrum of CO problems, incorporating more cutting-edge models, and integrating newer and more challenging distributions of instances.

While one might expect that extensively tailored heuristics can outperform learned approaches, our empirical findings suggest that simple local search heuristics frequently outperform complicated GNN-based heuristics. Specifically, TABU SEARCH, a local search heuristic that tries to avoid solutions previously encountered, outperforms all but one of the evaluated learned heuristics across a broad range of instance distributions. In addition, we show that ablating the GNN from ECO-DQN does not hurt its performance, and show evidence that ECO-DQN may simply be learning a heuristic similar to TABU SEARCH. On the positive side, ANYCSP did show a modest improvement over TABU SEARCH, although it uses many times the computational resources. Further, we observe that in some cases using a GNN to guide a traditional heuristic can improve the performance of the original heuristic.

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

## A    Appendix

### A.1    Detailed Description of Datasets

In this section, we provide details on the datasets, groups of datasets, and random graph models used in this paper.

- **Barrett et al. (2020).** This dataset consists of Erdős-Rényi and Barabási-Albert graphs (ER and BA respectively) with edge weights $w \in \{0, \pm 1\}$ and up to 500 vertices. We set the same parameters from the paper and generate ER graphs using $p = 0.15$ with edge weights $w \in \{0, \pm 1\}$ for training and validation. Similarly, for BA graphs , we use $m = 4$ with edge weights $w \in \{0, \pm 1\}$. For testing, we use 100 test graphs from this distribution from this dataset.
- **Erdős-Rényi (ER).** This well-known random graph model by Erdős et al. (1960) connects each pair of vertices with a probability $p$. Following Ye (2003), we set the same parameters from the paper and generate graphs of size $|V| = 800$ using $p = 0.06$ with edge weights $w \in \{0, \pm 1\}$ and $w \in \{0, 1\}$ for training and validation.
- **Barabási-Albert (BA).** This random graph model by Albert & Barabási (2002) iteratively adds nodes, connecting them to $m$ already existing nodes. For our experiments, we generate graphs of size $|V| = 800$ using $m = 4$, with edge weights $w \in \{0, \pm 1\}$ and $w \in \{0, 1\}$.
- **Holme-Kim (HK).** This random graph model by Holme & Kim (2002), similar to the BA model, includes an extra step for each randomly created edge that forms a triangle with probability $p$. We generate graphs of size $|V| = 800$ using $m = 4$ and $p = 0.10$, with edge weights $w \in \{0, \pm 1\}$ and $w \in \{0, 1\}$.
- **Watts-Strogatz (WS).** The random graph model by Watts & Strogatz (1998) starts with a well-structured ring lattice with a mean degree of $k$. In the next step, each edge is replaced with probability $p$ by another edge sampled uniformly at random. This approach aims to preserve "small-world properties" while maintaining a random structure similar to ER graphs. We generate graphs of size $|V| = 800$ using $k = 4$ and $p = 0.15$, with edge weights $w \in \{0, \pm 1\}$ and $w \in \{0, 1\}$.
- **Gset.** This dataset (Ye, 2003) is extensively used to benchmark classical heuristics (Benlic & Hao, 2013; Leleu et al., 2019; 2021) for MaxCut. The dataset comprises three types of weighted and unweighted random graphs: ER graphs with uniform edge probabilities, Planar graphs with decaying connectivity, and regular Toroidal graphs. For generating training and validation distributions, we use the independent graph generator Rudy by Giovanni Rinaldi, which is used for generating Gset graphs as sourced from Ye (2003). For training and validation, we generate ER, the union of Planar, and regular Toroidal graphs for training and validation. The arguments for generating the graphs with Rudy are collected from Helmberg & Rendl (2000).
- **Optsicom.** This distribution comes from a publicly available library of MaxCut instances[2] and includes synthetic and realistic instances that are widely used in the optimization community (see references on the library website). For training and validation, we generate a similar distribution, and for testing, we make use of a subset of the instances available, namely ten problems from Ising Spin glass models in physics (the first 10 instances in Set2 of the library) following Khalil et al. (2017). All ten instances have 125 nodes and 375 edges, with edge weights $w \in \{0, \pm 1\}$.
- **Sherrington-Kirkpatrick spin glass.** This distribution contains dense Sherrington-Kirkpatrick instances with elements $J_{ij} \in \{-1, 1\}$ generated from ER graphs based on examples from Hamerly et al. (2019). We generate graphs of size 70 to 100 vertices for training and validation. For testing, we use instances with the best-known value provided in CIM-Optimizer (Chen et al., 2022).
- **Phase transition.** This distribution contains dense unweighted instances from ER graphs at the phase transition ($p = 0.5$) (Coppersmith et al., 2004), based on examples from Hamerly et al. (2019). We generate graphs with 100 to 200 vertices for training and validation and make use of test instances provided in CIM-Optimizer (Chen et al., 2022).
- **Big Mac Library.** We select MaxCut instances from this dataset generated with Rudy, using edge weights $w \in \{0, \pm 1\}$. These instances were collected during the development of the Big Mac.
- **MaxCut and BQP Instance Library.** This distribution comes from a publicly available library of MaxCut instances[3]. For testing, we select Ising spin glass instances with edge weights $w \in \{0, \pm 1\}$ and vertices ranging from 1000 to 2744.

## A.2 Detailed Description of Benchmark Algorithms

All algorithms make use of 50 attempts, and the best solution found is reported, except for S2V, which is deterministic. Next, we provide details about each algorithm discussed in our paper.

---

[2]https://grafo.etsii.urjc.es/optsicom/index.php.html
[3]http://bqp.cs.uni-bonn.de/library/html/instances.html

**Mixed-Interger program solver.**

- **Gurobi.** GUROBI is a commercial mathematical optimization solver. In our experiments, we limit the number of threads to 10 and set a cut-off time of 10 seconds.
- **Cplex.** CPLEX is another popular optimization solver. We use similar settings as GUROBI for the experiments.

**Classical Heuristics.**

- **Greedy.** The algorithm starts with an empty solution and greedily moves the node which results in the largest improvement in cut weight until no greedy actions are left.
- **Semdifinite Programming (SDP).** SEMIDEFINITE PROGRAMMING is a subset of convex optimization that involves optimizing a linear objective function over the intersection of the cone of positive semidefinite matrices and an affine space. In our experiments, we use cvxpy (Diamond & Boyd, 2016) with default settings and limit the number of threads to 10 to ensure consistency with other methods.
- **Tabu Search (TS).** The algorithm begins with an arbitrary solution and, at each step, selects a node to flip (either including or excluding it from the solution set) that results in the greatest increase in objective value among the nodes that have not been selected for a specified hyper-parameter called Tabu Tenure. However, if selecting a node results in the best objective value found so far, TS disregards that rule. This algorithm continues searching for a fixed number of steps even when no further immediate improvements can be made, thereby enhancing the search process by exploring a broader solution space and potentially discovering better solutions over time. Various improved versions of this algorithm have been proposed Glover (1990); we consider the vanilla version of the algorithm (see Algorithm 1). Further details can be found in Glover (1990).
- **Extremal optimization (EO).** The algorithm begins with an initial arbitrary solution and sorts the vertices by their descending marginal gain. It then defines a probability distribution $P_k \propto k^{-\tau}$ where $1 \leq k \leq |V|$ for a given value of the parameter $\tau$ to determine the likelihood of selecting each vertex based on its rank in the sorted list. At each step, an index $k$ is sampled according to this probability distribution, and the membership of the selected vertex is flipped. This method allows the algorithm to escape local optima and explore the search space more effectively, thereby increasing the chances of finding better solutions. Similar to TS, it stops after a fixed number of iterations (see Algorithm 2). Further details can be found in Boettcher & Percus (2001).

**GNN-based heuristics**

- **S2V-DQN.** The algorithm starts from an empty solution and incrementally constructs solutions by adding one vertex at each step to the current solution, guided by a GNN . Once a vertex is added to the solution, it cannot be removed; in other words, the algorithm does not reverse its earlier decisions. The state space of its RL agent is represented by the current solution. The algorithm stops when no action can improve the objective value. The reward function of the algorithm is simply the change in the objective value. Further details can be found in Khalil et al. (2017).
- **ECO-DQN.** Unlike S2V-DQN, this algorithm starts with an arbitrary partition of vertices and allows reversible actions. Barrett et al. (2020) provides seven handcrafted features per node to represent the state space of its RL agent. At each step, it selects a vertex and flips its membership. The RL agent often chooses vertices that do not correspond to the greatest immediate increase in the objective value (non-greedy). Thus, it aims to strike a balance between exploitation and exploration of the search space. The algorithm stops after a fixed number of iterations. It provides a reward to the RL agent only when a new solution has been found, which equals the difference between the new best solution and the previous best solution. Since the reward can be very sparse, the algorithm also provides a small intermediate reward to the agent when the agent reaches a new locally minimal solution. Further details can be found in Barrett et al. (2020).
- **LS-DQN.** Similar to ECO-DQN, LS-DQN allows reversible actions and starts with an arbitrary solution instead of an empty one. The state space of its RL agent is represented by the current solution. At each step, it selects a vertex and flips the membership of the selected vertex. It stops after a fixed number of iterations or can terminate on its own. The reward function of this algorithm is defined as the negative

---

**Algorithm 1** Tabu Search

---

1: **Input:** Instance $G(V, E)$, Oracle $f$, Tabu Tenure $\gamma$, Maximum number of iterations $maxiter$
2: **Output:** $bestobj$
3: Initialize current solution $S \leftarrow$ random solution
4: Initialize tabu list $T$ as an empty dictionary
5: Initialize best objective value $bestobj \leftarrow f(S)$
6: Initialize iteration counter $iter \leftarrow 0$
7: **while** $iter < maxiter$ **do**
8:    $bestmove \leftarrow$ None
9:    $bestvalue \leftarrow -\infty$
10:    **for** each vertex $v \in V$ **do**
11:       Flip the membership of $v$ in $S$ to obtain $S'$
12:       Calculate objective value $f(S')$
13:       **if** $f(S') > bestobj$ **or** $(v \notin T$ **and** $f(S') > bestvalue)$ **then**
14:          $bestmove \leftarrow v$
15:          $bestvalue \leftarrow f(S')$
16:       **end**
17:    **end for**
18:    Flip the membership of $bestmove$ in $S$
19:    Insert or update $bestmove$ in tabu list $T$ with a value equal to the tabu tenure $\gamma$
20:    **if** $f(S) > bestobj$ **then**
21:       $bestobj \leftarrow f(S)$
22:    **end**
23:    **for** each vertex $v$ in tabu list $T$ **do**
24:       Decrease tabu tenure of $v$ by 1
25:       **if** tabu tenure of $v$ is 0 **then**
26:          Remove $v$ from $T$
27:       **end**
28:    **end for**
29:    Increment iteration counter $iter \leftarrow iter + 1$
30: **end while**
31: **return** $bestobj$

---

---

**Algorithm 2** Extremal Optimization Algorithm

---

**Input:** oracle $f$, graph $G(V, E)$, initial solution $S_0$, tau $\tau$, maximum iterations *maxiter*
 1: Initialize current solution $S \leftarrow S_0$
 2: Initialize a probability distribution $P_k \propto k^{-\tau}$ where $1 \leq k \leq |V|$
 3: Initialize best objective value $bestobj \leftarrow f(S)$
 4: Initialize iteration counter $iter \leftarrow 0$
 5: **while** $iter < maxiter$ **do**
 6:    Initialize a list $marginal\_gains \leftarrow []$
 7:    **for** each vertex $v \in V$ **do**
 8:       Flip the membership of $v$ in $S$ to obtain $S'$
 9:       Calculate objective value $f(S')$
10:       Calculate $gain \leftarrow f(S') - f(S)$
11:       Append $gain$ to $marginal\_gains$
12:    **end for**
13:    Sort vertices in descending order of marginal gains
14:    Select an index $k$ according to the probability distribution $P_k$
15:    Select the vertex $v_k$ that is in the $k$-th position in the sorted list
16:    Flip the membership of $v_k$ in $S$ to obtain new solution $S$
17:    **if** $f(S) > bestobj$ **then**
18:       $bestobj \leftarrow f(S)$
19:    **end**
20:    Increment iteration counter $iter \leftarrow iter + 1$
21: **end while**
22: **return** $bestobj$

---

value change of the objective function at each step. It generalizes to a variety of CO problems, like MaxCut and TSP. Further details can be found in Yao et al. (2021).

- **Gflow-CombOpt.** The algorithm begins with an empty solution. The formulation of the Markov decision process (MDP) for the generative flow network proceeds as follows: at each step, it adds one vertex to the solution. After each action, it checks if adding the vertex would decrease the cut value. If so, it excludes the vertex, ensuring it is never added back to the solution. Despite starting with an empty solution, the algorithm generates diverse solution candidates by sampling from a probability distribution in a sequential decision-making process. Further details can be found in Zhang et al. (2023).

- **RUN-CSP.** The algorithm solves CO problems that can be mapped to binary constraint satisfaction problems (CSP). It employs a graph neural network as a message-passing protocol, with the CSP instances modeled as a graph where nodes correspond to variables and edges represent constraints. Like other GNN-based heuristics, it is not a reinforcement learning approach; rather, the loss function to optimize this algorithm is designed to satisfy as many constraints as possible. The results show that it performs effectively on significantly larger instances, even when trained on relatively small ones. Further details can be found in Toenshoff et al. (2021).

- **ANYCSP.** The algorithm is an end-to-end search heuristic for any constraint satisfaction problem. Tönshoff et al. (2022) introduced a novel representation of CSP instances, called the constraint value graph, which allows for direct processing of any CSP instance. The state space of the RL agent is represented by both the current solution and the best solution found so far. At each step, the algorithm generates a soft assignment of variables within the CSP instance, enabling transitions between any two solutions in a single step. To encourage exploration and prevent the search from getting stuck in local maxima, a reward scheme similar to ECO-DQN is employed. Notably, the RL agent does not receive a reward upon reaching an unseen local minimum. Empirical evidence has shown that this approach can compete with or even surpass classical SOTA problem-specific heuristics. Further details can be found in Tönshoff et al. (2022).

**Quantum Annealing**

Quantum annealing algorithms start by framing the optimization problem as an energy landscape of a quantum system, with the solution being the state of the lowest energy. Initially, the quantum system is set in a superposition of all possible solutions, which represents a high-energy state. The objective is to steer the system towards the lowest energy state, which corresponds to the optimal or near-optimal solution for the problem. Next, we describe two SOTA quantum annealing algorithms used in our benchmark.

- **Amplitude Heterogeneity Correction (AHC).** The algorithm maps the objective function of CO problems to the energy landscape of a physical system called Coherent Ising Machine (Yamamoto et al., 2017). It relaxes the binary vertex states of the MaxCut problem to continuous values and finds low-energy states efficiently. It finds solutions of better or equal quality for GSet instances compared to those previously known from the classical SOTA heuristic Breakout Local Search (Benlic & Hao, 2013).
- **Chaotic Amplitude Control (CAC).** To improve the scalability, this algorithm makes use of non-relaxational dynamics that can accelerate the sampling of low-energy states to reduce the time to find optimal solutions. Further details can be found in Leleu et al. (2021).

### A.3 Baseline and Instance Bias

In this section, we continue our discussion on the lack of consensus regarding instances and baselines. This inconsistency leads to a situation where empirical results in different research papers are often not comparable. We present an overview of common baselines used to evaluate the performance of learned heuristics for MaxCut in Table 5. In Section 4.1, our empirical analysis shows that classical local search heuristics such as Tabu Search (Glover, 1990), Extremal Optimization (Boettcher & Percus, 2001) often outperform common baselines like Greedy, Semidefinite Programming (SDP) (Goemans & Williamson, 1995) and Cplex (IBM-ILOG, 2023) in practice. A comparison with these local search heuristics will help us gain insights into the performance improvements that learned heuristics offer.

Similarly, we observe from Table 6, that few heuristics actually evaluate their performance on hard instances that are used to benchmark SOTA heuristics.

### A.4 Reproducibility

In this section, we list all the changes we made to the previous implementations of algorithms and provide implementation details of the algorithms.

### A.4.1 S2V-DQN

We use the publicly available implementation of S2V-DQN by Barrett et al. (2020) as our initial codebase[4] for implementing S2V-DQN. This implementation had poor scalability because it used dense representations of graphs. It also could not handle distributions containing graphs of different sizes because dense representations of graphs of different sizes cannot be batched together. Therefore, we reimplement it to ensure it can scale up easily to larger instances. To ensure a fair comparison, we evaluate the performance of our implementation against pre-trained models provided by Barrett et al. (2022). From Figure 8, we observe that we have successfully reproduced the performance reported in the literature.

---

[4]Code available at: https://github.com/tomdbar/eco-dqn

Table 5: Overview of common baselines to evaluate the performance of learned heuristics.

| Baselines → | Operation Research | | Classical Heuristics | | | | Learned Heuristics | | |
|---|---|---|---|---|---|---|---|---|---|
| Approach ↓ | Cplex | Gurobi | SDP | Greedy | TS | EO | S2V-DQN | ECO-DQN | RUN-CSP |
| S2V-DQN | ✓ | | ✓ | ✓ | | | | | |
| ECO-DQN | ✓ | | | ✓ | | | ✓ | | |
| LS-DQN | | | ✓ | ✓ | | | ✓ | ✓ | |
| RUN-CSP | | | ✓ | ✓ | | ✓ | ✓ | | |
| ANYCSP | | | ✓ | ✓ | | | | ✓ | ✓ |
| Gflow-Comb | | ✓ | ✓ | ✓ | | | | | |

| Algorithms → Datasets ↓ | S2V-DQN | ECO-DQN | LS-DQN | RUN-CSP | ANY-CSP | GFLOW-COMBOPT |
|---|---|---|---|---|---|---|
| ER | ✓ | ✓ | | | | |
| BA | ✓ | ✓ | | | | ✓ |
| Regular | | | ✓ | ✓ | | |
| Gset | | ✓ | | ✓ | ✓ | ✓ |

Table 6: Overview of common datasets to evaluate the performance of learned heuristics.

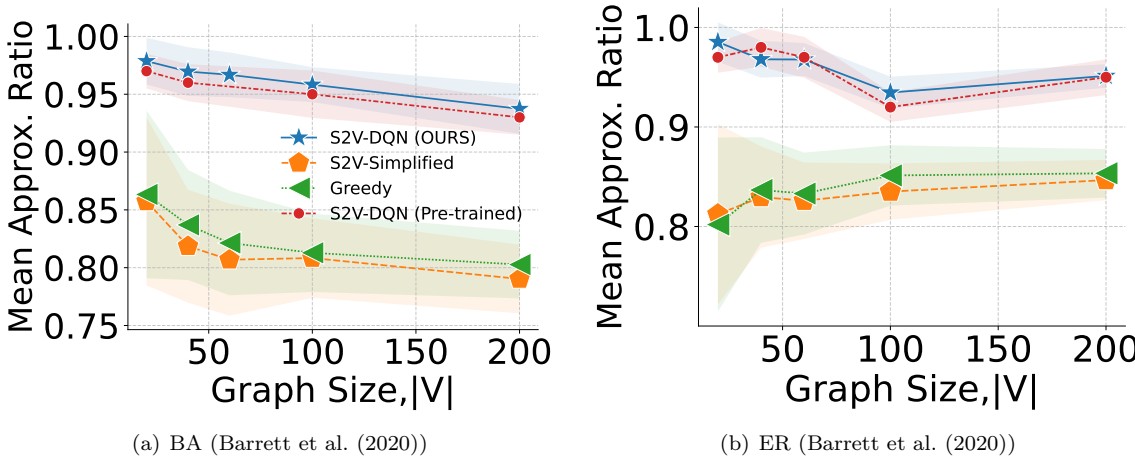

(a) BA (Barrett et al. (2020))

(b) ER (Barrett et al. (2020))

Figure 8: Performance comparison of S2V-DQN for reproducibility; Note that for BA graphs with 60 vertices, we were unable to load the pre-trained model for S2V-DQN.

### A.4.2 ECO-DQN

Similar to S2V-DQN, we use the publicly available implementation of ECO-DQN and make similar improvements as we did for S2V-DQN. To ensure reproducibility, we compare our implementation with the pre-trained networks provided by Barrett et al. (2020). From Figure 9, we observe that we have successfully reproduced the performance reported in the original paper.

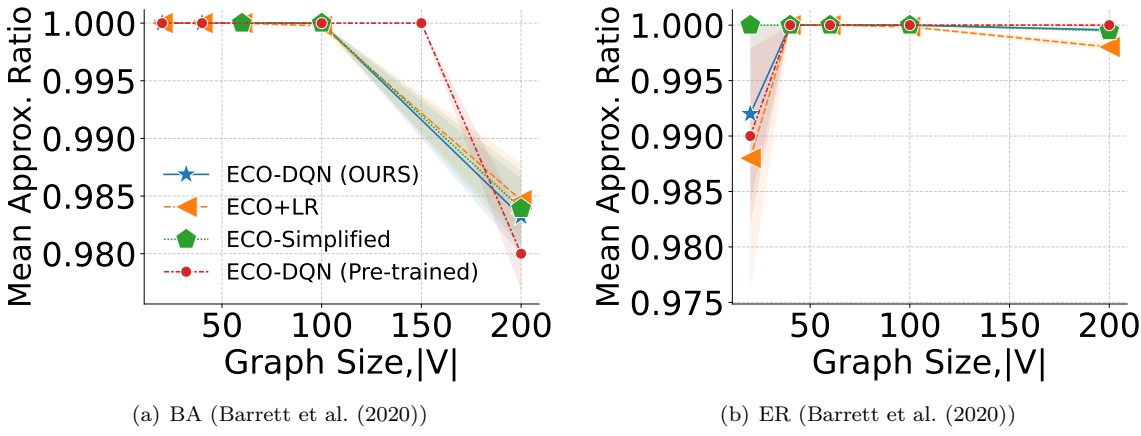

(a) BA (Barrett et al. (2020))

(b) ER (Barrett et al. (2020))

Figure 9: Performance comparison of ECO-DQN for reproducibility

Finally, we tested agents trained on weighted ER graphs with $|V| = 200$ on real-world datasets and hard instances, following the experimental setup of Barrett et al. (2020). From Table 7, we observe that ECO+LR outperforms ECO-DQN, except for ER graphs where ECO-DQN performs slightly better than ECO+LR.

Table 7: Average approximation ratios on known benchmarks: The second half of the table shows results for extended experiments.

| Dataset | Type | Nodes | ECO-Simplified | ECO+LR | ECO-DQN(Pre-trained) | ECO-DQN(OURS) |
|---------|------|-------|----------------|--------|----------------------|---------------|
| Physics | Regular | 125 | 1.000 | 1.000 | 1.000 | 1.000 |
| G1-10 | ER | 800 | 0.989 | 0.984 | 0.994 | 0.990 |
| G22-31 | ER | 2000 | 0.953 | 0.977 | 0.974 | 0.981 |

### A.4.3  LS-DQN

Since the LS-DQN codebase was not publicly available initially, we contacted the authors. Unfortunately, their codebase[5] is designed specifically for clustered graphs and is not suitable for training on arbitrary graph distributions. The authors did not provide details about  their configuration and hyper-parameter settings in the paper. However, we were able to replicate similar improvements over RG using the GNN, as reported in the paper (see Table 3).

### A.4.4  Gflow-CombOpt

For unweighted instances, we utilize the original implementation[6] of GFLOW-COMBOPT with the default configuration. For weighted instances, we incorporate weighted graph convolution to utilize edge weights and make necessary adjustments to ensure that a vertex is added to the solution set only if it improves the cut value compared to not including it. However, we observe poor empirical performance of the algorithm for weighted instances. Therefore, we restrict our empirical evaluation of GFlow-CombOPT to unweighted instances.

### A.4.5  RUN-CSP and ANYCSP

We use the PyTorch implementation of RUN-CSP[7] and ANYCSP[8] with the default configuration.

### A.4.6  CAC and AHC

We use the implementations of these two algorithms provided by Chen et al. (2022) and tune their hyper-parameters using Bayesian Optimization Hyperband.

### A.4.7  TS

To tune the value of tabu tenure, we run a grid search with a step size of 10, starting from 20 to 150, over the validation dataset and use the tuned tabu tenure for testing. We report the parameters used for our experiments in Table 8.

### A.4.8  EO

To tune the value of tau, we run a grid search with a step size of 0.1, starting from 1.1 to 1.9, over the validation dataset and use the tuned tau for testing. We report the parameters used for our experiments in Table 8.

### A.5  Additional results on generalization

Due to space constraints, we present the results of the generalization of learned and classical heuristics that are trained on a skewed graph distribution of size $|V| = 800$ from Gset and tested on various distributions of size $|V| = 2000$ from Gset. From Figure 10, we observe that simple heuristics match or outperform the performance of learned heuristics. In addition, we train all learned heuristics on ER graphs from Barrett et al. (2020) dataset and test on BigMac MaxCut, and BQP Instance Library. From Figure 11, we observe

---

[5]Code available at: https://github.com/MingzheWu418/LocalSearch-DQN
[6]Code available at: https://github.com/zdhNarsil/GFlowNet-CombOpt
[7]Code available at:https://github.com/toenshoff/RUNCSP-PyTorch
[8]Code available at:https://github.com/toenshoff/ANYCSP

Table 8: Parameters used for Tabu Search (TS) and Extremal Optimization (EO).

| Graph | Nodes | Tabu Tenure (TS) | Tau (EO) |
|---|---|---|---|
| Gset (ER) | 800 | 80 | 1.4 |
| GSet (Skew) | 800 | 90 | 1.4 |
| BA | 800 | 110 | 1.3 |
| WS | 800 | 140 | 1.4 |
| HK | 800 | 100 | 1.4 |
| Phase Transition | 100-200 | 20 | 1.8 |
| GSet (ER) | 800 | 100 | 1.7 |
| GSet (Skew) | 800 | 90 | 1.4 |
| GSet (Torodial) | 800 | 100 | 1.4 |
| BA | 800 | 120 | 1.2 |
| WS | 800 | 110 | 1.3 |
| HK | 800 | 110 | 1.2 |
| Barrett et al. (ER) | 200 | 10 | 1.9 |
| Barrett et al. (BA) | 200 | 20 | 1.6 |
| SK spin-glass | 70-100 | 20 | 1.8 |
| Physics (Regular) | 125 | 20 | 1.4 |

that ECO-DQN and SoftTabu (ECO+LR) achieve comparable performance, while S2V-DQN and RUN-CSP perform significantly worse.

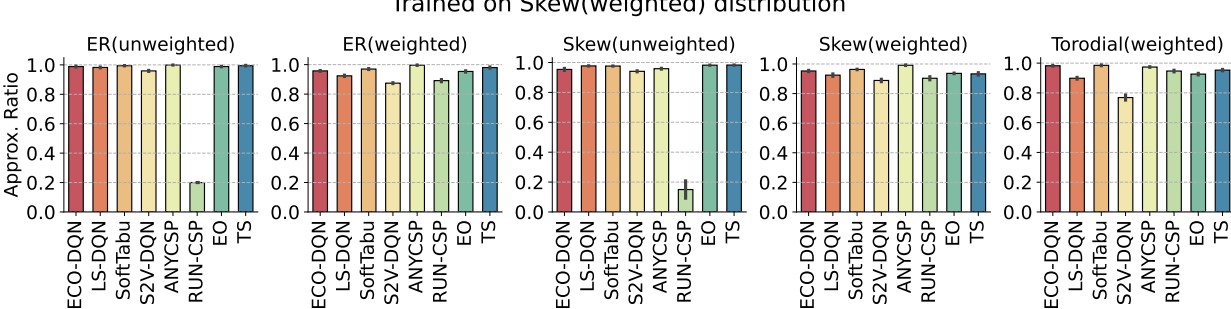

Figure 10: Generalization of agents to unseen graph sizes and structures.

## A.6 Additional details on Evaluation settings

Since we are using the previous codebase from several works, and they are not equally optimized, we use the number of search steps instead of a timeout. As both S2V-DQN and Gflow-CombOpt are irreversible (only add to the solution set), these algorithms can run for a maximum of $|V|$ steps. For other learned algorithms, we run the experiments for $4|V|$ steps and find no significant improvement in the objective value. We present the results of the learned heuristics in Table 9.

## A.7 Efficiency and Trade-offs in Large-Scale Instances

In this subsection, we present the run-time and mean approximation ratio of each algorithm evaluated across multiple datasets in Table 10.

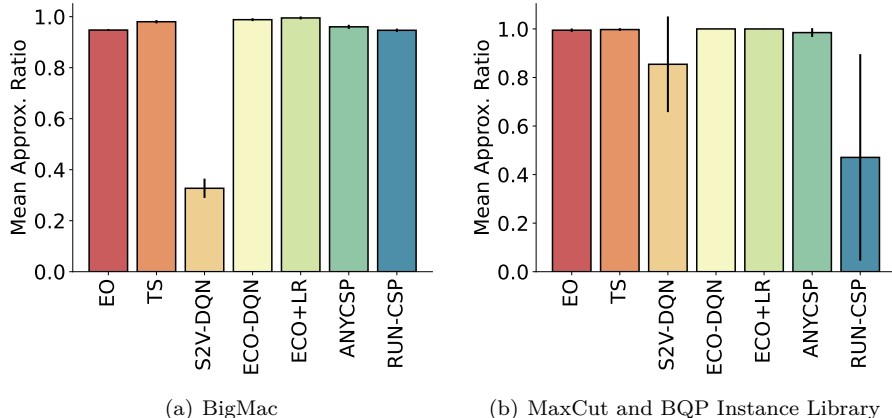

(a) BigMac

(b) MaxCut and BQP Instance Library

Figure 11: Additional experiments on generalizations on BigMac and MaxCut and BQP Instance Library.

Table 9: Average approximation ratios of learned heuristics optimized for $4|V|$ steps: The first and second halves of the table show results for unweighted and weighted instances, respectively.

| Graph | Nodes | ECO-DQN | LS-DQN | RUN-CSP | ANYCSP |
|---|---|---|---|---|---|
| Gset (ER) | 800 | $0.997^{\pm0.001}$ | $0.993^{\pm0.001}$ | $0.979^{\pm0.002}$ | $0.999^{\pm0.000}$ |
| GSet (Planar) | 800 | $0.989^{\pm0.001}$ | $0.983^{\pm0.002}$ | $0.954^{\pm0.001}$ | $0.997^{\pm0.001}$ |
| BA | 800 | $0.992^{\pm0.002}$ | $0.983^{\pm0.003}$ | $0.980^{\pm0.002}$ | $1.000^{\pm0.000}$ |
| WS | 800 | $0.992^{\pm0.002}$ | $0.972^{\pm0.003}$ | $0.979^{\pm0.003}$ | $1.000^{\pm0.000}$ |
| HK | 800 | $0.991^{\pm0.002}$ | $0.983^{\pm0.003}$ | $0.979^{\pm0.003}$ | $1.000^{\pm0.000}$ |
| Phase Transition | 100-200 | $1.000^{\pm0.000}$ | $0.998^{\pm0.001}$ | $0.984^{\pm0.005}$ | $1.000^{\pm0.000}$ |
| GSet (ER) | 800 | $0.981^{\pm0.006}$ | $0.950^{\pm0.008}$ | $0.912^{\pm0.009}$ | $0.998^{\pm0.002}$ |
| GSet (Planar) | 800 | $0.967^{\pm0.008}$ | $0.946^{\pm0.018}$ | $0.914^{\pm0.018}$ | $0.995^{\pm0.005}$ |
| GSet (Torodial) | 800 | $0.992^{\pm0.004}$ | $0.964^{\pm0.003}$ | $0.974^{\pm0.002}$ | $0.999^{\pm0.002}$ |
| BA | 800 | $0.973^{\pm0.008}$ | $0.933^{\pm0.010}$ | $0.937^{\pm0.009}$ | $1.000^{\pm0.000}$ |
| WS | 800 | $0.961^{\pm0.007}$ | $0.922^{\pm0.008}$ | $0.954^{\pm0.007}$ | $1.000^{\pm0.000}$ |
| HK | 800 | $0.975^{\pm0.009}$ | $0.937^{\pm0.009}$ | $0.944^{\pm0.007}$ | $1.000^{\pm0.000}$ |
| Barrett et al. (ER) | 200 | $1.000^{\pm0.001}$ | $0.988^{\pm0.008}$ | $0.945^{\pm0.012}$ | $1.000^{\pm0.000}$ |
| Barrett et al. (BA) | 200 | $0.983^{\pm0.031}$ | $0.977^{\pm0.032}$ | $0.960^{\pm0.016}$ | $0.989^{\pm0.037}$ |
| SK spin-glass | 70-100 | $1.000^{\pm0.000}$ | $0.999^{\pm0.002}$ | $0.962^{\pm0.019}$ | $1.000^{\pm0.001}$ |
| Physics (Regular) | 125 | $1.000^{\pm0.000}$ | $0.995^{\pm0.009}$ | $0.982^{\pm0.008}$ | $1.000^{\pm0.000}$ |

Table 10: A comparsion of run-time and performance of all learned and classical heristics across all datasets.

| | Greedy | EO | TS | S2V-DQN | LS-DQN | ECO-DQN | ECO+LR | Gflow | RUN-CSP | ANYCSP |
|---|---|---|---|---|---|---|---|---|---|---|
| BA200 (Weighted) | 0.892 (0.511) | 0.970 (1.827) | 0.969 (1.250) | 0.937 (0.538) | 0.978 (2.117) | 0.982 (3.024) | 0.984 (1.887) | NaN | 0.943 (1.151) | 0.986 (2.298) |
| BA800 (Unweighted) | 0.953 (0.447) | 0.982 (2.216) | 0.992 (1.766) | 0.950 (1.079) | 0.984 (20.208) | 0.991 (23.814) | 0.993 (10.579) | 0.955 (3.586) | 0.980 (14.529) | 1.000 (34.663) |
| BA800 (Weighted) | 0.859 (0.466) | 0.947 (2.344) | 0.976 (1.797) | 0.885 (1.190) | 0.960 (23.788) | 0.972 (25.524) | 0.978 (11.179) | NaN | 0.936 (14.880) | 1.000 (34.655) |
| ER200 (Weighted) | 0.946 (0.514) | 0.986 (1.837) | 0.990 (1.261) | 0.951 (0.659) | 0.988 (3.689) | 0.999 (4.508) | 0.998 (2.422) | NaN | 0.894 (4.376) | 1.000 (4.867) |
| ER800 (Unweighted) | 0.982 (0.436) | 0.994 (1.898) | 0.997 (1.523) | 0.970 (1.320) | 0.992 (62.094) | 0.985 (72.632) | 0.997 (23.740) | 0.979 (8.963) | 0.982 (63.578) | 0.999 (112.258) |
| ER800 (Weighted) | 0.915 (0.528) | 0.963 (2.442) | 0.986 (1.951) | 0.906 (1.637) | 0.947 (66.036) | 0.985 (72.546) | 0.987 (26.114) | NaN | 0.876 (66.641) | 0.998 (112.278) |
| HK800 (Unweighted) | 0.952 (0.447) | 0.982 (2.287) | 0.992 (1.831) | 0.966 (2.503) | 0.983 (20.560) | 0.991 (24.561) | 0.993 (11.448) | 0.930 (3.441) | 0.979 (14.510) | 1.000 (34.735) |
| HK800 (Weighted) | 0.858 (0.454) | 0.945 (2.356) | 0.971 (1.801) | 0.904 (1.125) | 0.947 (20.019) | 0.970 (24.693) | 0.975 (11.305) | NaN | 0.940 (3.722) | 0.996 (8.673) |
| Phase Transition (Unweighted) | 0.995 (0.469) | 0.999 (2.216) | 0.999 (1.687) | 0.985 (0.240) | 0.998 (6.379) | 1.000 (7.323) | 1.000 (3.984) | 0.924 (0.271) | 0.985 (5.117) | 1.000 (8.441) |
| Physics (Weighted) | 0.886 (0.519) | 0.996 (2.077) | 0.978 (1.257) | 0.962 (0.187) | 0.996 (1.189) | 1.000 (1.700) | 1.000 (1.147) | NaN | 0.989 (0.343) | 1.000 (0.922) |
| Planar800 (Unweighted) | 0.962 (0.525) | 0.985 (2.342) | 0.991 (1.862) | 0.980 (1.118) | 0.984 (24.553) | 0.989 (29.583) | 0.990 (14.206) | 0.967 (3.694) | 0.954 (19.079) | 0.997 (41.041) |
| Planar800 (Weighted) | 0.870 (0.595) | 0.938 (2.540) | 0.973 (1.930) | 0.890 (1.154) | 0.936 (26.087) | 0.967 (32.551) | 0.978 (15.048) | NaN | 0.914 (19.644) | 0.998 (41.531) |
| SK70-100 (Weighted) | 0.993 (0.516) | 0.993 (1.847) | 0.999 (1.746) | 0.939 (0.284) | 0.999 (2.164) | 1.000 (2.289) | 1.000 (1.347) | NaN | 0.884 (2.748) | 1.000 (2.464) |
| Torodial800 (Weighted) | 0.804 (0.615) | 0.947 (2.413) | 0.985 (1.911) | 0.960 (1.119) | 0.962 (15.014) | 0.993 (18.850) | 0.992 (9.013) | NaN | 0.975 (2.431) | 0.995 (6.580) |
| WS800 (Unweighted) | 0.927 (0.448) | 0.978 (2.223) | 0.989 (1.724) | 0.972 (1.064) | 0.975 (14.355) | 0.990 (18.403) | 0.986 (8.728) | 0.957 (2.943) | 0.978 (9.494) | 1.000 (27.113) |
| WS800 (Weighted) | 0.829 (0.682) | 0.939 (2.763) | 0.964 (1.788) | 0.915 (1.105) | 0.931 (16.339) | 0.957 (18.850) | 0.970 (9.009) | NaN | 0.951 (9.877) | 0.998 (27.125) |

## A.8 Additional Tables

Table 11: Performance of learned local search heuristics.

| Graph | Nodes | Classical Heuristics | | | | Learned Heuristics | | | |
|---|---|---|---|---|---|---|---|---|---|
| | | Greedy | TS | EO | S2V-Simplified | S2V-DQN | LS-DQN | ECO-DQN | ECO+LR |
| ER (Unweighted) | 800 | $0.984^{\pm0.001}$ | $0.998^{\pm0.001}$ | $0.994^{\pm0.001}$ | $0.97^{\pm0.003}$ | $0.97^{\pm0.001}$ | $0.992^{\pm0.001}$ | $0.984^{\pm0.002}$ | $0.997^{\pm0.001}$ |
| | 1000 | $0.971^{\pm0.003}$ | $0.996^{\pm0.001}$ | $0.99^{\pm0.001}$ | $0.959^{\pm0.003}$ | $0.947^{\pm0.003}$ | $0.968^{\pm0.002}$ | $0.972^{\pm0.002}$ | $0.995^{\pm0.002}$ |
| | 2000 | $0.966^{\pm0.001}$ | $0.994^{\pm0.001}$ | $0.988^{\pm0.001}$ | $0.959^{\pm0.004}$ | $0.945^{\pm0.001}$ | $0.963^{\pm0.002}$ | $0.967^{\pm0.001}$ | $0.994^{\pm0.001}$ |
| | 5000 | $0.922^{\pm0.0}$ | $0.965^{\pm0.0}$ | $0.97^{\pm0.0}$ | $0.932^{\pm0.0}$ | $0.924^{\pm0.0}$ | $0.838^{\pm0.0}$ | $0.918^{\pm0.0}$ | $0.977^{\pm0.0}$ |
| | 7000 | $0.92^{\pm0.0}$ | $0.962^{\pm0.0}$ | $0.964^{\pm0.0}$ | $0.931^{\pm0.0}$ | $0.918^{\pm0.0}$ | $0.834^{\pm0.0}$ | $0.916^{\pm0.0}$ | $0.974^{\pm0.0}$ |
| | 10000 | $0.901^{\pm0.0}$ | $0.92^{\pm0.0}$ | $0.932^{\pm0.0}$ | $0.931^{\pm0.0}$ | $0.873^{\pm0.0}$ | $0.828^{\pm0.0}$ | $0.896^{\pm0.0}$ | $0.96^{\pm0.0}$ |
| ER (Weighted) | 800 | $0.919^{\pm0.011}$ | $0.986^{\pm0.002}$ | $0.965^{\pm0.007}$ | $0.834^{\pm0.03}$ | $0.906^{\pm0.015}$ | $0.95^{\pm0.013}$ | $0.981^{\pm0.005}$ | $0.988^{\pm0.005}$ |
| | 2000 | $0.87^{\pm0.007}$ | $0.977^{\pm0.003}$ | $0.95^{\pm0.004}$ | $0.82^{\pm0.011}$ | $0.869^{\pm0.007}$ | $0.852^{\pm0.008}$ | $0.947^{\pm0.005}$ | $0.976^{\pm0.003}$ |
| | 5000 | $0.809^{\pm0.0}$ | $0.91^{\pm0.0}$ | $0.912^{\pm0.0}$ | $0.781^{\pm0.0}$ | $0.806^{\pm0.0}$ | $0.43^{\pm0.0}$ | $0.923^{\pm0.0}$ | $0.942^{\pm0.0}$ |
| | 7000 | $0.803^{\pm0.0}$ | $0.91^{\pm0.0}$ | $0.91^{\pm0.0}$ | $0.784^{\pm0.0}$ | $0.819^{\pm0.0}$ | $0.436^{\pm0.0}$ | $0.924^{\pm0.0}$ | $0.945^{\pm0.0}$ |
| Toroidal (Weighted) | 800 | $0.798^{\pm0.012}$ | $0.982^{\pm0.003}$ | $0.942^{\pm0.004}$ | $0.728^{\pm0.02}$ | $0.96^{\pm0.009}$ | $0.964^{\pm0.003}$ | $0.992^{\pm0.004}$ | $0.993^{\pm0.003}$ |
| | 2000 | $0.766^{\pm0.003}$ | $0.955^{\pm0.004}$ | $0.926^{\pm0.001}$ | $0.734^{\pm0.011}$ | $0.958^{\pm0.006}$ | $0.95^{\pm0.003}$ | $0.984^{\pm0.002}$ | $0.987^{\pm0.001}$ |
| | 5000 | $0.759^{\pm0.0}$ | $0.897^{\pm0.0}$ | $0.905^{\pm0.0}$ | $0.726^{\pm0.0}$ | $0.954^{\pm0.0}$ | $0.937^{\pm0.0}$ | $0.976^{\pm0.0}$ | $0.979^{\pm0.0}$ |
| | 7000 | $0.757^{\pm0.0}$ | $0.909^{\pm0.0}$ | $0.9^{\pm0.0}$ | $0.731^{\pm0.0}$ | $0.954^{\pm0.0}$ | $0.943^{\pm0.0}$ | $0.978^{\pm0.0}$ | $0.979^{\pm0.0}$ |
| | 10000 | $0.755^{\pm0.003}$ | $0.864^{\pm0.012}$ | $0.89^{\pm0.002}$ | $0.731^{\pm0.001}$ | $0.954^{\pm0.002}$ | $0.939^{\pm0.001}$ | $0.975^{\pm0.0}$ | $0.979^{\pm0.0}$ |
| Planar (Unweighted) | 800 | $0.964^{\pm0.003}$ | $0.99^{\pm0.001}$ | $0.986^{\pm0.002}$ | $0.96^{\pm0.001}$ | $0.98^{\pm0.001}$ | $0.984^{\pm0.001}$ | $0.99^{\pm0.001}$ | $0.99^{\pm0.001}$ |
| | 1000 | $0.962^{\pm0.001}$ | $0.99^{\pm0.001}$ | $0.985^{\pm0.001}$ | $0.957^{\pm0.003}$ | $0.978^{\pm0.001}$ | $0.984^{\pm0.0}$ | $0.989^{\pm0.001}$ | $0.99^{\pm0.001}$ |
| | 2000 | $0.959^{\pm0.0}$ | $0.983^{\pm0.001}$ | $0.982^{\pm0.001}$ | $0.956^{\pm0.003}$ | $0.981^{\pm0.001}$ | $0.979^{\pm0.0}$ | $0.983^{\pm0.002}$ | $0.985^{\pm0.002}$ |
| | 5000 | $0.956^{\pm0.0}$ | $0.979^{\pm0.0}$ | $0.98^{\pm0.0}$ | $0.956^{\pm0.0}$ | $0.972^{\pm0.0}$ | $0.775^{\pm0.0}$ | $0.976^{\pm0.0}$ | $0.98^{\pm0.0}$ |
| | 7000 | $0.958^{\pm0.0}$ | $0.98^{\pm0.0}$ | $0.98^{\pm0.0}$ | $0.957^{\pm0.0}$ | $0.972^{\pm0.0}$ | $0.773^{\pm0.0}$ | $0.975^{\pm0.0}$ | $0.978^{\pm0.0}$ |
| Planar (Weighted) | 800 | $0.868^{\pm0.005}$ | $0.977^{\pm0.005}$ | $0.949^{\pm0.008}$ | $0.816^{\pm0.022}$ | $0.89^{\pm0.023}$ | $0.943^{\pm0.006}$ | $0.965^{\pm0.002}$ | $0.978^{\pm0.003}$ |
| | 2000 | $0.86^{\pm0.003}$ | $0.944^{\pm0.004}$ | $0.933^{\pm0.001}$ | $0.809^{\pm0.023}$ | $0.889^{\pm0.007}$ | $0.923^{\pm0.007}$ | $0.947^{\pm0.005}$ | $0.961^{\pm0.003}$ |
| | 5000 | $0.845^{\pm0.0}$ | $0.925^{\pm0.0}$ | $0.925^{\pm0.0}$ | $0.82^{\pm0.0}$ | $0.889^{\pm0.0}$ | $0.904^{\pm0.0}$ | $0.931^{\pm0.0}$ | $0.956^{\pm0.0}$ |
| | 7000 | $0.845^{\pm0.0}$ | $0.918^{\pm0.0}$ | $0.918^{\pm0.0}$ | $0.815^{\pm0.0}$ | $0.882^{\pm0.0}$ | $0.894^{\pm0.0}$ | $0.937^{\pm0.0}$ | $0.956^{\pm0.0}$ |

Table 12: T-test results between different methods.

| Comparison | t-statistic | p-value | Significance |
|---|---|---|---|
| TS & ECO+LR | 0.218 | 0.829 | No |
| TS & ECO+GNN | 0.602 | 0.551 | No |
| ECO+LR & ECO+GNN | 0.425 | 0.674 | No |

