# OpenReview forum: "MaxCutBench: Revisiting and Benchmarking Graph Neural Networks for Maximum Cut"
_TMLR — Accepted by TMLR_

### Review · Reviewer_xnth · 2025-02-15

**Summary Of Contributions:**

This paper seeks to benchmark a bunch of GNNs and other methods for the MaxCut problem, i.e., finding a subset $S$ of the vertices of a weighted graph $G=(V,E,w)$ such that $\sum_{v\in S,w\notin S} w(u,v)$ is as large as possible. Towards this end, they contribute an open source code base and compare several classical heuristics with more modern methods.  In their experiments, they find several surprising and interesting results.

My opinion is that this work will be suitable for publication in TMLR, but only after the requested changes are made.

**Audience:**

Yes

**Claims And Evidence:**

Yes

**Requested Changes:**

Please address the run-time issues discussed in ``weaknesses" as well as charts being hard to read. (If you can address the other two weaknesses, that would be great, but not required.)

For Quantum Annealing, it says that you report the best of the two solvers. It would be better to report both results in the interest of transparency (possibly in appendix if there are spatial constraints).

Minor:

Page 3, Start of Section 3 Should write G=(V,E,w) rather than G(V,E) since th graph is weighted

Page 3, ``section 4.2" should be capitalized (since it is a proper noun). Please check throughout.

**Strengths And Weaknesses:**

Strengths:

Many of the findings are interested and some of them are surprising. For instance that classical results are outperforming GNNs; that replacing GNN-components with Linear regression does not lead to loss of performance; and that GFlowNets perform so poorly in the context of weighted graphs.

Weaknesses:

It is unclear how generalizable these findings would be to CO problems more broadly, beyond MaxCut.

Work is purely empirical. In an ideal world, it would be nice having some theory relating the statistical properties of the network / generating model to the performance of various algorithms.

Unclear how the performance of the algorithms is related to run-time. There is some exploration of computational efficiency in Section 4.5, but this needs to be further developed by a table directly showing run-time and performance together. Is there a trade-off where the more accurate algorithms come with a longer run-time.

Some of the performance charts are hard to read. Please also include the results in tabular form. If needed, these can be in the appendix, but they should be mentioned in the main body, e.g., ``results are also shown in Table 7 in Appendix A"

---

> ### Author Response · Authors · 2025-03-09
> **Response to Reviewer xnth**
>
> Thank you for your thoughtful review and valuable feedback. We sincerely appreciate the time and effort you have taken to assess our work.
>
> ## Weakness
>
> > - It is unclear how generalizable these findings would be to CO problems more broadly, beyond MaxCut.
> > - Work is purely empirical. In an ideal world, it would be nice having some theory relating the statistical properties of the network / generating model to the performance of various algorithms.
> >  - Unclear how the performance of the algorithms is related to run-time. There is some exploration of computational efficiency in Section 4.5, but this needs to be further developed by a table directly showing run-time and performance together. Is there a trade-off where the more accurate algorithms come with a longer run-time.
> > - Some of the performance charts are hard to read. Please also include the results in tabular form. If needed, these can be in the appendix, but they should be mentioned in the main body, e.g., ``results are also shown in Table 7 in Appendix A"
>
> As mentioned in the requested change, we will address runtime and readability issues in our revised manuscript.
>
> ## Requested Change
>
> > Please address the run-time issues discussed in ``weaknesses" as well as charts being hard to read. (If you can address the other two weaknesses, that would be great, but not required.)
>
> We will carefully address the runtime issue and improve the readability of the charts in our final manuscript.
>
> > For Quantum Annealing, it says that you report the best of the two solvers. It would be better to report both results in the interest of transparency (possibly in appendix if there are spatial constraints).
>
> We appreciate this suggestion and will include both results for transparency, ensuring they are available either in the main text or the appendix, depending on space constraints.
>
> > Minor:
> > - Page 3, Start of Section 3 Should write G=(V,E,w) rather than G(V,E) since th graph is weighted
> > - Page 3, ``section 4.2" should be capitalized (since it is a proper noun). Please check throughout
>
> Thank you for pointing these out. We will carefully make these corrections and review the manuscript for consistency.

---

### Review · Reviewer_r9Re · 2025-03-01

**Summary Of Contributions:**

Summary: This paper rethinks the baselines of the current GNN-based learning method for solving the Maximum Cut problem. This paper selects several recently proposed/highly cited GNN-based methods (S2V-DQN, ECO-DQN, ANYCSP, and Gflow-CombOpt) and classical heuristics (such as Tabu Search and Greedy), and compares them on several frequently used datasets. The results show that the GNN module does not provide an effect on performance for some GNN-based methods in many cases. The simple local search heuristics (such as Tabu Search) frequently outperform complicated GNN-based heuristics. However, the  ANYCSP (a learning method) does perform well.

Contribution: 1) This paper proposes MaxCut-Bench, a benchmark suite that provides a fair and comparable environment for evaluations of GNN-based learning methods on the Maximum Cut problem. 2) Using the proposed MaxCut-Bench, this paper reveals that local search heuristics such as Tabu Search can outperform most learning methods. 3) this paper also reveals that the GNN in some GNN-based methods may not play a role in performance.

**Audience:**

Yes

**Claims And Evidence:**

Yes

**Requested Changes:**

none

**Strengths And Weaknesses:**

Strengths:
1) The motivation is clear, and the experimental settings are reasonable in most cases.
2) This writing is clear and easy to understand.
3) The conclusion is somehow interesting and may give new insight into this community.

Weaknesses/Questions:
1) Does S2V-DQN is a constructive method? If so, it may be unfair for S2V-DQN to compare with the local search method Tabu Search. In other words, S2V-DQN should be compared with the heuristic Greedy. The conclusion that S2V-DQN is beaten by Tabu Search may not be sound. Further, refining the categories of the methods will be better.
2) Why ANYCSP performs so well? This paper describes the phenomenon but does not give further insights.
3) In ECO-DQN's origin paper, does it ablate the effect of GNN? If yes, what is the conclusion in their paper? Do they also find the uselessness of GNN?
4) Is there any other recently proposed learning-based method for solving the Maximum Cut problem? For example, the methods that are published in 2024.

---

> ### Author Response · Authors · 2025-03-09
> **Response to Reviewer r9Re**
>
> Thank you for your thoughtful review and insightful comments. We greatly appreciate your time and effort in evaluating our work.
>
> > 1. Does S2V-DQN is a constructive method? If so, it may be unfair for S2V-DQN to compare with the local search method Tabu Search. In other words, S2V-DQN should be compared with the heuristic Greedy. The conclusion that S2V-DQN is beaten by Tabu Search may not be sound. Further, refining the categories of the methods will be better.
>
> Thank you for raising this point. Yes, S2V-DQN is a constructive method. For a fair comparison, we have also included the iterative version of S2V-DQN, LS-DQN. Both the constructive and iterative versions of S2V-DQN can be outperformed by Tabu Search.
>
> > 2. Why ANYCSP performs so well? This paper describes the phenomenon but does not give further insights.
>
> While we attempted to understand why ANYCSP performs well, we did not gain many insights from the algorithm. Upon revisiting the original paper, the authors attribute the performance improvement to the global search action space, which allows modifying any number of variables at each step of the stochastic search.
>
> > 3. In ECO-DQN's origin paper, does it ablate the effect of GNN? If yes, what is the conclusion in their paper? Do they also find the uselessness of GNN?
>
> In the original paper, they did not ablate the effect of the GNN. One takeaway from our paper is that ablation of major components, such as the GNN, should be considered.
>
> > 4. Is there any other recently proposed learning-based method for solving the Maximum Cut problem? For example, the methods that are published in 2024.
>
> Not to the best of our knowledge, without a potentially expensive polynomial-time reduction from another problem to MaxCut. We chose these learning-based methods, as they had a high impact and are general-purpose and had already been evaluated on MaxCut in the original paper.

---

### Review · Reviewer_949y · 2025-03-01

**Summary Of Contributions:**

The paper introduces MaxCutBench, an extensive benchmark suite for evaluating algorithms on the Maximum Cut (MaxCut) problem. This benchmark addresses inconsistencies in evaluation practices for comparing both classical and deep learning-based heuristics. The paper highlights the lack of strong baselines in the research community, which limits the ability to measure real improvements in neural network-based approximation methods.

**Audience:**

Yes

**Claims And Evidence:**

Yes

**Requested Changes:**

## Questions / Clarifications about experiments:
- How are CPLEX and Gurobi tuned for hyperparameters?
   - These solvers perform extra work to prove optimality, so it should be clearly stated that they belong to a different class of optimization methods.
   - Using default parameters for both solvers may not be fair; just as heuristics are hand-tuned, solver parameters should be tuned for fair comparisons.

- Are deep learning heuristics retrained on each problem set?
    - This is assumed but not explicitly mentioned in the Evaluation section.
    - Is S2V-DQN retrained for each plot in Figure 2?

- Experimental setup for Section 4.4 is unclear.
    - How are out-of-distribution (OOD) graphs selected?
    - Are deep learning heuristics retrained for each dataset?

## Minor comments
- Figure 2: Legends are poorly placed and should be adjusted for clarity.
- Figure captions: Should be more descriptive, summarizing key findings instead of just stating the experiment type.
- Table 2: "Summery" → "Summary" (Typo).

**Strengths And Weaknesses:**

## Strengths

- The paper proposes a benchmark to address performance evaluation inconsistencies in the research community on maximum cut problem families.
- The authors present an extensive collection of problem sets and evaluate a diverse range of algorithms.
- The experimental setup provides insights into performance characteristics of classical and learned methods.


## Weaknesses

### Comments on experiments
- The metric of time to solve is missing from all the tables and plots, making it unclear how computational efficiency compares across methods. Ideally, there should be a plot of time on the x-axis and a number of problems solved on the y-axis to give a measure of computational efficiency of a method. Ignore this point, if time budget is kept constant across the methods and please mention it explicitly in the experiments.

- In Section 4.1, CPLEX and Gurobi are not tuned, although they belong to the family of exact optimization solvers. It is unclear why a 10s cutoff was chosen.

- Inconsistent naming conventions for baseline algorithms make interpretation difficult. For example:
    - Greedy is renamed LS-Simplified, giving the impression that they are different methods.
    - Tabu Search (TS) is called ECO-Simplified in Section 4.2, which is misleading.

- I do not completely agree with the conclusions regarding ECO-DQN in Section 4.2:
     - p-values are missing to show whether differences between methods are statistically significant.
     - Table 3 shows significant overlap in confidence bands for ECO-LR and ECO-GNN, suggesting their performance is nearly identical. This point can be raised in a separate section to highlight the need for proper ablations.
     - The claim that ECO-DQN holds dominance over TS is questionable without additional statistical tests. It seems both are equally winning across the problem families, and the claim that TS is superior doesn’t sound valid.
     - The relative standing of EO against ECO-DQN is not explicitly discussed.

- The inclusion of S2V-DQN and LS-DQN in Section 4.2 is unclear. The only meaningful comparison appears to be ECO-DQN.
- Section 4.3 should have been the focal point, as it directly addresses the paper’s research question. However, the comparisons in this section are not extensive enough.
     - All of the DQN methods are missing.
     - Quantum Annealing is mentioned but not used anywhere in the paper
     - This section should have included the performance on all the problem families across all the methods.

- Section 4.5 lacks a detailed discussion of factors affecting memory and efficiency.
      - The observed memory usage might be an artifact of implementation rather than an inherent limitation of the algorithm.
      - Time to solve should have been one of the metric in previous sections anyway.



### Broader comments

- The paper appears to be discussing two distinct issues:
     - The lack of a unified evaluation criterion in the MaxCut research community.
     - The presentation of MaxCutBench as a new benchmark to standardize evaluation.

- The focus is heavily skewed toward the first point, while the second point is not convincingly presented. To make MaxCutBench more compelling, the authors should:
     *  Show performance of all methods across all datasets in a single visualization.
     *  Provide a clearer justification for why MaxCutBench should be adopted as a standard benchmark. Other than providing an unified evaluation, what are the pros and cons of using this benchmark? What insights does this benchmark reveal about where the community currently lacks progress? Based on these findings, what are the key open challenges and future research directions?


- Additionally, the authors could mention the role of proof-of-concepts (POCs) and iterative refinements in algorithmic research. For example:
    - S2V-DQN can be viewed as a POC, which, despite not outperforming all baselines, introduced a new paradigm for solving MaxCut.
While iterative refinements have since improved upon it, such improvements might be evaluated relative to the original POC rather than meaningful real-world progress.
    - In contrast, MaxCutBench provides an extensive benchmark where improvements are directly compared against state-of-the-art (SOTA) traditional heuristics, ensuring that advances are substantive rather than incremental.

- Sections 4.1, 4.2, and 4.3 could be consolidated into a single section for better readability and presentation of results.

---

> ### Author Response · Authors · 2025-03-09
> **Response to Reviewer 949y**
>
> We thank the reviewer for their comments and feedback. We hope these answers address the major concerns, particularly the empirical contribution. Please let us know if further questions or clarifications are needed.
>
> ## Comments on experiments
> > The metric of time to solve is missing from all the tables and plots, making it unclear how computational efficiency compares across methods. Ideally, there should be a plot of time on the x-axis and a number of problems solved on the y-axis to give a measure of computational efficiency of a method. Ignore this point, if time budget is kept constant across the methods and please mention it explicitly in the experiments.
>
> We agree that the time budget of each experiment should be more clear. In Section 4.5, we compare the time to solution of the various methods. The methods are allowed different amounts of time because they require a wide range of durations to find the best solutions, and a fixed time budget would not adequately capture the trade-off between time and optimality.
> We agree that a combined figure would better illustrate the computational efficiency of each method, and will create one for the next version of the manuscript.
> > In Section 4.1, CPLEX and Gurobi are not tuned, although they belong to the family of exact optimization solvers. It is unclear why a 10s cutoff was chosen.
>
> We experimented with different time budgets and empirically found that, within a 10-second cutoff, Gurobi can often find the publicly available best-known solutions or solutions very close to them (as shown in Figure 1), whereas CPLEX takes much longer to find good solutions and typically performs worse as the problem size grows. Similar results are reported in [1]. Even with a 10-minute cutoff, CPLEX performs very poorly on medium-sized graphs (|V| ≥ 200). Similar results are reported in [2]. Hence, we select 10 seconds as our cutoff to reflect that Gurobi is a much faster baseline with good performance compared to CPLEX, making it a better reference for comparison, even though both belong to the same family of exact optimization solvers.
>
> > Inconsistent naming conventions for baseline algorithms make interpretation difficult. For example: Greedy is renamed LS-Simplified, giving the impression that they are different methods and Tabu Search (TS) is called ECO-Simplified in Section 4.2, which is misleading.
>
> We agree that our naming convention is confusing. We were trying to draw attention to which algorithm the corresponding deep learning approach most closely resembles. However, in the next version, we will adopt a uniform naming convention for each algorithm.
>
> > p-values are missing to show whether differences between methods are statistically significant.
>
> T-test between TS and ECO+LR: t-statistic=0.218, p-value=0.829
>
> T-test between TS and ECO+GNN: t-statistic=0.602, p-value=0.551
>
> T-test between ECO+LR and ECO+GNN: t-statistic=0.425, p-value=0.674
>
> Statistical Significance (alpha = 0.05):
>
> TS vs ECO+LR: Not significant
>
> TS vs ECO+GNN: Not significant
>
> ECO+LR vs ECO+GNN: Not significant
>
> We agree that the differences between the methods are not statistically significant. As mentioned in our paper (Page 8), we conclude that there is no significant difference in performance between ECO-DQN and its simpler counterparts (TS and ECO-LR). Adding the GNN adversely affects the generalization performance of ECO-DQN, as discussed in Section 4.4.
>
> > Table 3 shows significant overlap in confidence bands for ECO-LR and ECO-GNN, suggesting their performance is nearly identical. This point can be raised in a separate section to highlight the need for proper ablations.
>
> We also address this in the final considerations of Section 4.3 and will revise our manuscript to emphasize it further.
>
> > The claim that ECO-DQN holds dominance over TS is questionable without additional statistical tests. It seems both are equally winning across the problem families, and the claim that TS is superior doesn’t sound valid.
>
> We completely agree with this statement, as mentioned in our paper (page 8).
>
> > The relative standing of EO against ECO-DQN is not explicitly discussed.
>
> We will revise our final manuscript to address this issue.
>
>
> [1] Barrett, Thomas, et al. "Exploratory combinatorial optimization with reinforcement learning." Proceedings of the AAAI conference on artificial intelligence. Vol. 34. No. 04. 2020.
>
> [2] Khalil, Elias, et al. "Learning combinatorial optimization algorithms over graphs." Advances in neural information processing systems 30 (2017).

---

> ### Author Response · Authors · 2025-03-09
> **Response to Reviewer 949y**
>
> ## Comments on experiments (Continuing)
>
> > The inclusion of S2V-DQN and LS-DQN in Section 4.2 is unclear. The only meaningful comparison appears to be ECO-DQN.
>
> The main purpose of including S2V-DQN and LS-DQN is to confirm that the performance improvement claims over their simpler counterparts in the original manuscript hold for hard instances, validate our successful reproduction of these works, and demonstrate that while deep learning can enhance local search, it can still be easily outperformed by simple handcrafted heuristics that improve upon local search. Moreover, we show that LS-DQN performs worse than ECO-DQN on hard instances, contrary to the claim made in the original paper, highlighting the need for testing on the hard instances.
>
> > Section 4.3 should have been the focal point, as it directly addresses the paper’s research question. However, the comparisons in this section are not extensive enough.
> > - All of the DQN methods are missing.
>
> We compare all DQN methods in Section 4.1 and observe that TS predominantly outperforms all DQN methods. Therefore, for brevity, we compare the remaining GNN-based heuristics with TS and EO.
>
> > Quantum Annealing is mentioned but not used anywhere in the paper
>
> The purpose of quantum annealing algorithms is to find the best solution rather than directly comparing them with other algorithms, so we set a high cutoff time of 1 hour. All approximation ratios reported in the paper are based on the best solution found by either of the two quantum annealing solvers.
>
> > This section should have included the performance on all the problem families across all the methods.
>
> We will revise our final manuscript to address this issue.
>
>
> > Section 4.5 lacks a detailed discussion of factors affecting memory and efficiency. - The observed memory usage might be an artifact of implementation rather than an inherent limitation of the algorithm. - Time to solve should have been one of the metric in previous sections anyway.
>
> We implement algorithms using efficient graph learning packages designed for scalability, enabling the handling of larger instances. However, it is difficult to argue that a more efficient solution does not exist. We will merge our analysis of time with previous sections for better readability.
>
> ## Broader comments
>
> > The focus is heavily skewed toward the first point, while the second point is not convincingly presented. To make MaxCutBench more compelling, the authors should:
> > - Show performance of all methods across all datasets in a single visualization.
>
> We will include a figure summarizing all the results in our final manuscript.
>
> > Provide a clearer justification for why MaxCutBench should be adopted as a standard benchmark. Other than providing an unified evaluation, what are the pros and cons of using this benchmark? What insights does this benchmark reveal about where the community currently lacks progress? Based on these findings, what are the key open challenges and future research directions?
>
>
> The main goal of our work is to provide a fair and comparable environment for evaluating machine learning algorithms for combinatorial optimization, specifically for the maximum cut problem. In our view, our work highlights the lack of standardization in datasets and baselines needed to demonstrate significant improvements. It is our hope that MaxCutBench could fill this need.
>
> In addition to our negative results, we also find that classic algorithms, such as local search, can be significantly enhanced with deep learning [1,2]. This suggests that more research is needed on replacing handcrafted rules with machine learning and standardization in datasets and baselines.
>
>
>
>
> [1] Barrett, Thomas, et al. "Exploratory combinatorial optimization with reinforcement learning." Proceedings of the AAAI conference on artificial intelligence. Vol. 34. No. 04. 2020.
>
> [2] Khalil, Elias, et al. "Learning combinatorial optimization algorithms over graphs." Advances in neural information processing systems 30 (2017).

---

> ### Author Response · Authors · 2025-03-09
> **Response to Reviewer 949y**
>
> ## Boarder Comments (Continuing)
>
> > Additionally, the authors could mention the role of proof-of-concepts (POCs) and iterative refinements in algorithmic research. For example:
> > - S2V-DQN can be viewed as a POC, which, despite not outperforming all baselines, introduced a new paradigm for solving MaxCut. While iterative refinements have since improved upon it, such improvements might be evaluated relative to the original POC rather than meaningful real-world progress.
> > - In contrast, MaxCutBench provides an extensive benchmark where improvements are directly compared against state-of-the-art (SOTA) traditional heuristics, ensuring that advances are substantive rather than incremental.
>
> Thank you for pointing this out. We agree that this is a good viewpoint and discussion to add to the next version of the manuscript.
>
> > - Sections 4.1, 4.2, and 4.3 could be consolidated into a single section for better readability and presentation of results.
>
> While we appreciate the suggestion, we believe keeping Sections 4.1, 4.2, and 4.3 separate helps maintain clarity and structure in presenting the results. Each section focuses on a distinct aspect, and consolidating them might make it harder for readers to follow the specific details of each analysis. However, we are open to suggestions on improving transitions between sections to enhance readability.
>
> ##  Questions / Clarifications about experiments:
>
> > How are CPLEX and Gurobi tuned for hyperparameters?
> > - These solvers perform extra work to prove optimality, so it should be clearly stated that they belong to a different class of optimization method.
> > - Using default parameters for both solvers may not be fair; just as heuristics are hand-tuned, solver parameters should be tuned for fair comparisons.
>
> Thank you for pointing this out, as the other algorithms do not do this extra work to prove optimality. We will add a short discussion of this to the text. We experimented with different time budgets and empirically found that, within a 10-second cutoff, Gurobi can often find the publicly available best-known solutions or solutions very close to them (as shown in Figure 1), whereas CPLEX takes much longer to find good solutions and typically performs worse as the problem size grows. Similar results are reported in [1]. Even with a 10-minute cutoff, CPLEX performs very poorly on medium-sized graphs (|V| ≥ 200). Similar results are reported in [3]. Hence, we select 10 seconds as our cutoff to reflect that Gurobi is a much faster baseline with good performance compared to CPLEX, making it a better reference for comparison, even though both belong to the same family of exact optimization solvers.
>
> > Are deep learning heuristics retrained on each problem set?
> > - This is assumed but not explicitly mentioned in the Evaluation section.
>
> Yes, the deep learning methods are retrained for each problem set.
>
> > Experimental setup for Section 4.4 is unclear.
> > - How are out-of-distribution (OOD) graphs selected?
> > - Are deep learning heuristics retrained for each dataset?
>
> - We select the Gset dataset for evaluating the generalization performance of learning-based heuristics following [1] and [2].
> - We train on three different datasets and evaluate their performance on out-of-distribution datasets.
>
> > Minor comments
> > - Figure 2: Legends are poorly placed and should be adjusted for clarity.
> > - Figure captions: Should be more descriptive, summarizing key findings instead of just stating the experiment type.
> > - Table 2: "Summery" → "Summary" (Typo).
>
> We will address these issues in our final manuscript.
>
>
>
>
> [1] Barrett, Thomas, et al. "Exploratory combinatorial optimization with reinforcement learning." Proceedings of the AAAI conference on artificial intelligence. Vol. 34. No. 04. 2020.
>
> [2] Tönshoff, Jan, et al. "One model, any CSP: graph neural networks as fast global search heuristics for constraint satisfaction." arXiv preprint arXiv:2208.10227 (2022).
>
> [3] Khalil, Elias, et al. "Learning combinatorial optimization algorithms over graphs." Advances in neural information processing systems 30 (2017).
>
> [4] Yao, Fan, Renqin Cai, and Hongning Wang. "Reversible action design for combinatorial optimization with reinforcement learning." arXiv preprint arXiv:2102.07210 (2021).

---

> > ### Comment · Reviewer_949y · 2025-03-12
> > **thank you**
> >
> > I have gone through the comments. The response addresses most of my concerns. Thanks for agreeing to reflect the changes in the new manuscript.
> >
> > - Regarding the time, it’s a standard practice to compare methods across time and the quality of the solution - often done through survival plots. For example, ANYCSP also compares algorithms on the time vs solved plot (figure 3). Section 4.5 might not mean a lot without such comparison. Consider a scenario where the problem is too complex to solve for ANYCSP, in which case it takes 100 times more time to solve than it takes on other issues. This single outlier can skew the aggregate time ot solve the problem, whereas ANYCSP might be solving the rest of the problem faster than the different methods.
> >
> >
> > - Regarding tuning CPLEX or Gurobi, my question was whether CPLEX’s parameters can be tuned to yield better performance. Time limit is just one of those parameters, but there are more parameters that are related to heuristics, cut selection, etc.
> >
> >
> > - I think its a good practice to include p-values, so it will be useful to have them in the final version.
> >
> >
> > - Regarding the comparison in Section 4.3, I think it will be helpful to keep the comparison consistent -  switching to objective value instead of keeping approximation ratio on y-axis makes it confusing to compare across sections. At the same time, perhaps including ECO-DQN (best among the previous methods), could be included in the plot. And even better would be to have a survival plot as well.
> >
> >
> > - Finally, I think the manuscript can benefit from a dedicated paragraph on the role of POCs (as authors already agreed), giving credit to the previous works.

---

> > > ### Comment · Reviewer_949y · 2025-03-13
> > > **one more question -**
> > >
> > > One more question worth clarifying: Since Gurobi is already solving optimally, why is Quantum Anealing required? Shouldn't the approximation ratios computed using Gurobi's solution?

---

> > > > ### Author Response · Authors · 2025-03-14
> > > > **Response to one more question by Reviewer 949y**
> > > >
> > > > > One more question worth clarifying: Since Gurobi is already solving optimally, why is Quantum Anealing required? Shouldn't the approximation ratios computed using Gurobi's solution?
> > > >
> > > > Gurobi with 10s cutoff is not achieving the optimal solution, so we normalize by QA following [1]. However, which algorithm we normalize by doesn’t affect the conclusions since we primarily focus on classical handcrafted heuristics and learning-based heuristics.
> > > >
> > > > [1] Barrett, Thomas, et al. "Exploratory combinatorial optimization with reinforcement learning." Proceedings of the AAAI conference on artificial intelligence. Vol. 34. No. 04. 2020.
> > > >
> > > > We will upload our revised manuscript by March 15th. Thank you for your thoughtful feedback. Please let us know if you have any further questions or suggestions.

---

> > > ### Author Response · Authors · 2025-03-14
> > > **Response to Reviewer 949y**
> > >
> > > Thank you for your thoughtful review and insightful comments. We greatly appreciate your time and effort in evaluating our work.
> > >
> > > > Regarding the time, it’s a standard practice to compare methods across time and the quality of the solution - often done through survival plots. For example, ANYCSP also compares algorithms on the time vs solved plot (figure 3). Section 4.5 might not mean a lot without such comparison. Consider a scenario where the problem is too complex to solve for ANYCSP, in which case it takes 100 times more time to solve than it takes on other issues. This single outlier can skew the aggregate time ot solve the problem, whereas ANYCSP might be solving the rest of the problem faster than the different methods.
> > >
> > > Thank you for the suggestion regarding the comparison of methods across time and quality of solutions. We will revise Section 4.5 to include a time vs mean approximation ratio plot, as we believe this will provide a better comparison.
> > >
> > >
> > > > Regarding tuning CPLEX or Gurobi, my question was whether CPLEX’s parameters can be tuned to yield better performance. Time limit is just one of those parameters, but there are more parameters that are related to heuristics, cut selection, etc.
> > >
> > > Regarding the tuning of CPLEX, we used the default parameters following [1,2]. We agree that tuning CPLEX could have a significant impact, but there are many such parameters and such a tuning is expensive and nontrivial.
> > >
> > > >I think its a good practice to include p-values, so it will be useful to have them in the final version.
> > >
> > > We will include p-values in the final version, as we agree that they would provide useful statistical context to our findings.
> > >
> > > >Regarding the comparison in Section 4.3, I think it will be helpful to keep the comparison consistent - switching to objective value instead of keeping approximation ratio on y-axis makes it confusing to compare across sections. At the same time, perhaps including ECO-DQN (best among the previous methods), could be included in the plot. And even better would be to have a survival plot as well.
> > >
> > > We will revise the plot to maintain the approximation ratio on the y-axis rather than switching to objective value. Additionally, we will include ECO-DQN as the best method among previous approaches.
> > >
> > > > Finally, I think the manuscript can benefit from a dedicated paragraph on the role of POCs (as authors already agreed), giving credit to the previous works.
> > >
> > > We appreciate the suggestion to dedicate a paragraph to the role of POCs and give credit to previous works. We will include this in the final manuscript to ensure proper acknowledgment.
> > >
> > > [1] Barrett, Thomas, et al. "Exploratory combinatorial optimization with reinforcement learning." Proceedings of the AAAI conference on artificial intelligence. Vol. 34. No. 04. 2020.
> > >
> > > [2] Khalil, Elias, et al. "Learning combinatorial optimization algorithms over graphs." Advances in neural information processing systems 30 (2017).

---

### Comment · Action_Editor_mnsn · 2025-02-22

Dear Authors,

Thank you very much for submitting your work to TMLR. I am writing to provide you with an update on the review process.

As of now, we have only received one review by the original deadline (February 21). Due to unforeseen circumstances, one reviewer has requested an extension and will submit their review within the next 10 days. Additionally, as we have not received any response from the other reviewer, we have recruited a new emergent reviewer to ensure a thorough evaluation of your work.

With these adjustments, I expect to have at least three reviews completed within the next two weeks (by March 8). I appreciate your understanding of this arrangement.

Please feel free to reach out if you have any questions or concerns.

Best Regards,

AE

---

### Comment · Action_Editor_mnsn · 2025-03-01

Dear Authors,

We have now received three reviews for your paper: two from the initial reviewers and one from the emergent reviewer. In accordance with TMLR guidelines, I have unassigned the unresponsive reviewer from this submission. As a result, no additional review comments will be provided.

Please ensure that your rebuttal carefully addresses the weaknesses and requested changes raised in the three available reviews.

Best Regards,

AE

---

### Author Response · Authors · 2025-03-09
**General Comment**

We would like to thank all of the reviewers again for helping us improve the paper. We will upload a revised version of our paper as soon as possible.

---

### Author Response · Authors · 2025-03-15
**Updated manuscript**

We sincerely thank all the reviewers for their valuable feedback, which has helped us improve the paper. We have uploaded a revised version, addressing all your suggestions. Specifically, we:

- Justified why **MaxCutBench** should be adopted as a standard benchmark in the introduction section.
- Summarized the performance of all methods across datasets (Figure 1) and added a mean approximation ratio vs. time analysis in Section 4.5 to highlight the trade-off between performance and efficiency.
- Performed statistical tests comparing ECO-DQN with its simpler counterparts to assess statistical significance and included the results in our paper.
- Emphasized the importance of ablation studies and proof-of-concept in Section 4.2.
- Expanded the discussion on all methods, analyzing the relative standing of EO against ECO-DQN in Section 4.3. We also updated Figure 5 to include ECO-DQN and ensured consistency by reporting approximation ratios.
- Reported run-time and performance metrics for each algorithm across datasets in Appendix 10.
- Fixed typos and inconsistent naming conventions, improved figure captions, and added tables to enhance readability of performance charts.
- Due to the large number of graphs, we will upload results on which QA algorithm performs best to our GitHub repository.

---

### Decision · Action_Editor_mnsn · 2025-03-22

**Recommendation:** Accept as is

**Comment:**

The reviewers found the proposed MaxCutBench useful for the research community with clear motivation, and the experimental findings were interesting with valuable insights. While the reviewers identified several weaknesses and requested changes in their comments, most of these concerns were adequately addressed during the rebuttal. In the official recommendation, two reviewers vote to accept this work, and one reviewer leans toward acceptance.

I read the paper in detail and fully agree with the reviewers' consensus that this work clearly satisfies the TMLR acceptance criteria (solid and well-supported claims, potential audience). I therefore recommend accepting this work.

In the camera-ready version, please ensure that all promised changes are meticulously incorporated into the paper.

**Audience:**

All reviewers believe some individuals in TMLR's audience could be interested in the findings of this paper.

**Claims And Evidence:**

This work proposes MaxCutBench, a comprehensive benchmark suite for the Maximum Cut problem, which contains 16 traditional and learning-based algorithms with a unified interface. The main contributions are twofold: 1) the development of MaxCutBench itself, which includes modern re-implementations of widely-used learning-based heuristics, which offers a standardized framework for evaluation and comparison;2) a thorough analysis of the implemented learning-based algorithms and traditional algorithms for MaxCut, which reveals numerous interesting and surprising results and insights.

All reviewers believe the claims made in this paper are supported by accurate, convincing and clear evidence.